# Development and Evaluation of an Improved Off-Line Aerosol Mass Spectrometry Technique

Christina N. Vasilakopoulou[1,2], Kalliopi Florou[2], Christos Kaltsonoudis[2], Iasonas Stavroulas[3], Nikolaos Mihalopoulos[3] and Spyros N. Pandis[1,2]

[1]Department of Chemical Engineering, University of Patras, Patras, Greece
[2]Institute of Chemical Engineering Sciences, ICE-HT, Patras, Greece
[3]Institute for Environmental Research and Sustainable Development, National Observatory of Athens, Athens, Greece

*Correspondence to*: Spyros N. Pandis (spyros@chemeng.upatras.gr)

**Abstract.** The off-line Aerosol Mass Spectrometry (AMS) technique is a useful tool for the source apportionment of organic
aerosol (OA) in areas and periods during which an AMS is not available. However, the technique is based on the extraction of
aerosol samples in water, while several atmospheric OA components are partially or fully insoluble in water. In this work an
improved off-line technique was developed and evaluated in an effort to capture most of the partially soluble and insoluble
organic aerosol material, reducing significantly the uncertainty of the corresponding source apportionment. A major advantage
of the proposed approach is that no corrections are needed for the off-line analysis to account for the limited water solubility
of some OA components. The improved off-line AMS analysis was tested in three campaigns: two during winter and one
during summer. Collocated on-line AMS measurements were performed for the evaluation of the off-line method. Source
apportionment analysis was performed separately for the on-line and the off-line measurements using Positive Matrix
Factorization (PMF). The PMF results showed that the fractional contribution of each factor to the total OA differed between
the on-line and the off-line PMF results by less than 15%. The differences in the AMS spectra of the factors of the two
approaches could be significant suggesting that the use of factor profiles from the literature in the off-line analysis may lead
to complications. Part of the good agreement between the on-line and the off-line PMF results is due to the ability of the
improved off-line AMS technique to capture a bigger part of the OA, including insoluble organic material. This was evident
by the significant fraction of submicrometer suspended insoluble particles present in the water extract, and by the reduced
insoluble material on the filters after the extraction process. More than half of the elemental carbon (EC) was on average
missing from the filters after the water extraction. Significant EC concentrations were measured in the produced aerosol that
was used as input to the AMS during the off-line analysis.

## 1 Introduction

The High-Resolution Time-of-Flight Aerosol Mass Spectrometer (HR-ToF-AMS) has been used during the last fifteen years
for the source apportionment of organic aerosol (OA) in field studies (DeCarlo et al., 2008; Sun et al., 2011). The corresponding
results have dramatically improved our understanding of particulate matter and especially of organic aerosol. AMS is one of

the few instruments which can provide information for the size distribution of aerosols, their concentration, and their chemical composition in high temporal resolution (Drewnick et al., 2005; Jayne et al., 2000).

The AMS measurements can also be combined with source apportionment algorithms, such as Positive Matrix Factorization (PMF) (Paatero and Tapper, 1994) to estimate the contribution of the various sources or atmospheric transformations of OA. The advantage of PMF compared to other receptor models is that the solutions are constrained to be non-negative, which makes it suitable for environmental applications. The multilinear engine algorithm (ME-2) can be also used in cases that PMF results in non-meaningful solutions or to quantify smaller contributions of sources (Paatero, 1999). The difference between PMF and ME-2, is that in the latter, the user can use a-priori information about the factor profiles as input to the algorithm forcing it to account for the specific source.

Even though the AMS has been used in many field campaigns around the world (Aiken et al., 2009; Saarikoski et al., 2012; Setyan et al., 2012) there are circumstances in which its use is impractical. Its weight, size, and power consumption make sometimes its move to the field challenging or even impossible for some sites. Also, its high cost makes its use in multiple locations in the same city or region impossible. The off-line AMS technique, developed by Daellenbach et al. (2016) is a possible solution to both these problems. In the off-line analysis the ambient particulate matter is collected in filters. Then the filter samples are extracted in ultrapure water, and the water extract is filtered, aerosolized, dried, and finally measured with the AMS in the laboratory. Even though this technique provides valuable information about the OA, it has been designed to capture the water-soluble part of the measured OA. The ME-2 was proposed by Daellenbach et al. (2016) for the source apportionment of the corresponding off-line measurements. Comparisons between the on-line and the off-line measurements by Daellenbach et al. (2016) suggested the need for significant corrections for hydrocarbon-like OA (HOA), cooking OA (COA), and even biomass burning OA (BBOA) to account for the partial recovery of the components. Many studies have used the corrections proposed by Daellenbach et al. (2016) for OA to correct the off-line results for this partial recovery (Bozzetti et al., 2017; Vlachou et al., 2018; Srivastava et al., 2021). However, these corrections may introduce significant uncertainties in the approach (Ge et al., 2017). They may be location specific (Xu et al., 2017) or they may be sensitive to small variations in the experimental method. Some studies do not use these correction factors (Sun et al., 2011; Mihara and Mohinda, 2011; Xu et al., 2015; Ye et al., 2017; Li et al., 2021) and assume that the off-line results correspond only to the water-soluble part and not the total OA.

The application of the off-line technique may be limited by the OA mass loading of the filter. Niedek et al. (2022) developed a technique in order to reduce the detection limits of the off-line AMS measurements. Micronebulization was performed with only 10 μL of liquid used for the extraction. Organics and inorganics were quantitated by an isotopically labelled internal standard.

A significant difference between the on-line and the off-line measurements is the reduced temporal resolution of the latter. Typically, the on-line measurements are conducted every few minutes, in contrast to the filter collection which is in most cases performed daily. The effect of the reduced temporal resolution to the PMF results was addressed by Vasilakopoulou et al. (2022), who analyzed with PMF a five-month period in high (30 min) and low (daily) temporal resolution. The results

showed that the average contribution of each source to the total OA differed by less than 15% between the high and the low-resolution analysis. However, significant differences were observed in the factor spectra. These differences were of secondary importance for the results of the offline AMS source apportionment, however, they suggest that the use of the on-line factor spectra in the off-line analysis may lead to significant errors.

In this study we aim to improve the original off-line experimental procedure of Daellenbach et al. (2016) in order to
capture a higher fraction of the OA and reduce significantly the uncertainty of the PMF results. A major advantage of the original approach is that it is quantitative, however it requires a specific atomization procedure and does not account directly for the insoluble material. It relies on the use of correction factors that may or may not be applicable for a certain area. The new method is easier to apply in different laboratories and the analysis of its results is relatively straightforward because it directly accounts for the insoluble material. The proposed improved off-line AMS technique is evaluated on three field
campaigns in Greece, two during winter and one in summer. Comparisons with the on-line source apportionment results are used to evaluate the accuracy of the off-line source apportionment solution.

## 2 Methodology

### 2.1 Experimental

The proposed off-line method is based on the work of Daellenbach et al. (2016) but with changes in the extraction and
atomization steps in an effort to characterise most of the OA. Particulate matter was collected daily (24 h sampling) in quartz filters by a high-volume sampler (Tisch Scientific, 200 L min$^{-1}$). PM$_{2.5}$ cyclones were used for the aerosol sampling. After the collection of each sample, the filters were wrapped in pre-baked aluminium foil and placed in petri dishes. Subsequently they were placed in a freezer at -18$^{\circ}$C. Also, the samples were transferred frozen to the laboratory in order to reduce as much as possible the volatilization of semi-volatile organic compounds from the filters.

In the laboratory, filter punches (1.5 cm$^2$) are extracted in 20 mL of ultrapure water, with the help of an ultrasonic sonicator (Elmasonic S80). The extracts are placed in a syringe pump working at a flow rate of 15 mL h$^{-1}$. The water sample is not filtered in an effort to analyse a higher fraction of the OA and to limit the losses of organic aerosol material. This is one of the major features of our proposed approach. The extract is then atomized using an atomizer (TSI, 3075) and the produced droplets are dried using a silica gel dryer. The resulting aerosol is characterized by a HR-ToF-AMS (Aerodyne Inc.). The AMS
measures particles smaller than 1 μm because larger particles (e.g. fragments of the quartz filter) produced during the extraction and atomization processes are unable to pass through the AMS aerodynamic lenses. A Zetasizer (Malvern Nano ZS) was used to measure the size distribution of particles in the water extracts of the filters.

One of the potential problems of the technique is the fragmentation of the quartz filter during the sonication and the fate of the filter debris. In order to address this issue, we used the Zetasizer to measure the size distribution of particles in water
extracts of sonicated clean pre-baked filters and we compared the results with experiments in which we just sonicated clean water. In both cases, there were few particles in the sub-micrometer range and the filter presence made practically no

difference. However, in the case of the clean filter, a small peak was observed for particles with sizes around 5 μm. This is probably due to fragments of the quartz filter. However, these larger particles do not make it into the AMS (they do not pass through its aerodynamic lens) and therefore do not affect our measurements. Their presence did not cause any problems in the operation of our atomizer for the hundreds of samples that we have analyzed so far. As a quality assurance measure, we always compare the OA mass spectrum in the beginning and in the end of the off-line measurement and we have not seen any change. If something goes wrong with the atomization during a measurement, it would be probably detected this way. The atomizer is frequently (every few samples) cleaned to minimize any potential contamination.

On average around 40 min of off-line measurements are performed with the AMS for each collected sample. The temporal resolution of the off-line AMS measurements was 3 min. Before and after each measurement, a blank measurement is conducted using atomized the same ultra-pure water that is used for the extraction of the samples. The ultrapure water is atomized, the produced droplets are dried and the resulting aerosol is measured by the AMS, similarly with the actual sample, for around 30 minutes. The average of the two blank measurements is subtracted from the sample value. These blanks were measured during the same days when the actual samples were analyzed so they account for the small variations in the quality of the ultrapure water. We have also tested the blank correction performing the full procedure with a clean filter. The results were for all practical purposes the same as those for the clean water (the angle of the spectra was just 3 degrees). It should be also noted that particles too small to enter the AMS will necessarily have a very low mass concentration, so even if this material is added to the sample it will not affect our results. We have tested this by just looking at the size distributions of the produced aerosol from the blank experiments with an SMPS.

The proposed technique was evaluated in three different campaigns in Greece. The number of samples used in each period was above 30. In two of the campaigns (winter 2020 and early summer 2019 in Patras) an HR-AMS was used for the on-line measurements and in the third (winter 2019 in Athens) an Aerosol Mass Speciation Monitor (ACSM) (Aerodyne Inc., USA) was used.

## 2.2 On-line source apportionment

The PMF analysis of the on-line measurements (on-line PMF from now on) was performed using SoFi Pro in resolution of 3 min for the AMS and 30 min for the ACSM. No a-priori information about the factor profiles was used. OA high resolution (HR) mass spectra were analyzed ($m/z$ 12-300) (Canonaco et al., 2013) for the case of the AMS datasets. The "weak" signals (signal to noise (S/N) ratio was between 0.2 and 1) were down weighted by a factor of 2 and the "bad" (S/N below 0.2) by a factor of 10. Also, the variables related to $CO_2$ and $CO_2$ based corrections (16, 17, 18, 28 and 44) were down weighted by a factor of 2. The rotational ambiguity of the solution was explored using the $F_{peak}$. The minimum $F_{peak}$ value was -1, the maximum 1 and the step was 0.1.

For the Athens campaign the unit mass resolution (UMR) spectra ($m/z$ 12-125) were used, and they were pretreated in accordance to the procedure described above. Because of the low temporal resolution of the measurements (30 min) high concentration events were not removed.

## 2.3 Off-line source apportionment

The off-line PMF was also performed using SoFi Pro, without utilization of any a-priori information about the factor profiles. The high-resolution MS data were used for the off-line PMF in all three datasets. Each sampling day was represented by an off-line spectrum that was the arithmetic average of the 40 min off-line AMS data. The error for each sample was represented by the arithmetic average of the error of the off-line measurements. The measured AMS spectrum was corrected based on the blank measurements. The blank concentration was subtracted from the sample for each $m/z$ separately. The error matrix of the AMS spectrum was also blank-corrected using the same procedure. The blank correction was performed prior to the PMF analysis.

The same down weighting of weak and bad signals, and the same $F_{peak}$ approach as in the on-line measurements was used.

## 3 Field Campaigns

The improved off-line AMS technique was applied to more than 100 ambient daily filter samples from two urban areas in Greece and during different periods of the year. Continuous online AMS measurements were also performed, and they were used for the evaluation of the approach. In this work we will focus on the results from two winter periods (Athens and Patra) and one summer period in Greece (Patra). The measurements from each campaign are analysed separately in the following sections. For the two campaigns performed in Patras (winter 2020 and summer 2019) an HR-AMS, with 3 min temporal resolution was used, while in the Athens campaign an ACSM was used. The temporal resolution of the ACSM was 30 min.

### 3.1 Winter campaign in Patras, 2020

Patras is the third biggest city of Greece, with almost 200 thousand inhabitants. The winter campaign was conducted at the University of the Peloponnese campus during January-February 2020. This urban background site is located around 4 km from the centre of Patras. The main goal of the campaign was to study the biomass burning emissions and especially residential wood burning.

Both ambient measurements and mobile smog chamber experiments were conducted (Jorga et al., 2021) but in this study we focus on the ambient measurements. Together with several other instruments the HR-ToF-AMS was used to characterize the non-refractory $PM_1$ concentration and composition continuously for one month. The V-mode was used, and the temporal resolution of the measurements was 3 min. The vaporizer temperature was 600°C and no drying of the sample was performed. Collocated filter samples were also collected during the same period. The sampling started at 18:00 LT each day and lasted for 24 h. The black carbon (BC) mass concentration was measured using a Multi Angle Absorption Photometer (MAAP).

High PM$_1$ concentrations were measured during the late afternoon and night hours when biomass burning for heating purposes was taking place. In some periods the OA mass concentration exceeded 100 μg m$^{-3}$ (Fig. 1) with the organics accounting for around 70% of the total. Black carbon levels were also high during these periods exceeding 10 μg m$^{-3}$.

## 3.2 Summer campaign in Patras, 2019

The summer campaign in Patras was conducted in the Institute of Chemical Engineering Sciences (ICE-HT), which is located in a suburb of Patras, 8 km away from the city centre. The area is surrounded by olive tree fields and there is limited anthropogenic activity within a radius of 1 km.

The campaign lasted from March until June 2019. On-line AMS measurements took place during a few days every week, and mostly during weekends, due to the use of the AMS in other laboratory experiments during that period. Measurements were performed during 10 days per month on average. Filter samples were collected during the same days. Low PM$_1$ concentrations were observed during this measurement period (Fig.2). The aerosol levels were on average 4 μg m$^{-3}$. Among the four months, March was the one with the highest average OA concentration (6.9 μg m$^{-3}$) and May had the lowest (2 μg m$^{-3}$).

## 3.3 Winter campaign in Athens, 2019

Athens is the biggest city of Greece, with a population of 3.2 million inhabitants. The winter campaign was performed at the National Observatory of Athens at Thissio, in the centre of the city. The measurements started in January and lasted until March 2019. The OA in this case was measured using an Aerosol Mass Speciation Monitor (ACSM) (Aerodyne Inc., USA) with a temporal resolution of 30 min.

The OA was on average 8.2 μg m$^{-3}$ during the examined period (Fig. 3). The highest average concentration (10.8 μg m$^{-3}$) was observed in February, and the lowest (4.4 μg m$^{-3}$) in January. High concentration events were observed during the whole period late at night. The maximum half-hour OA concentration (98 μg m$^{-3}$) was observed on February 19 at midnight. The OA concentration started to increase from 22:00 LT on February 18 and remained high until 4:00 LT that night.

## 4 Source apportionment results

### 4.1 Source apportionment of the Patras 2020 winter campaign

#### 4.1.1 On-line

The one- up to six-factor solutions were explored, and the five-factor solution was chosen as the one that could explain satisfactorily the OA sources. Biomass burning (BBOA) was the dominant source of the OA (Fig. 1). Two biomass burning factors were identified (responsible for 53% of the total OA), a cooking OA (COA) factor (12%), a hydrocarbon-like OA (HOA) (10%) and an oxidized OA (OOA) factor (25%) (Fig. 1).

BBOA I and BBOA II had similar time series ($R^2$=0.91), but different spectra. BBOA I had strong peaks at *m/z* values 43, 60 and 73 (Ng et al., 2011). On the other hand, BBOA II was characterized by m/z's 44 and 60 (Figs. S3-S4). The two BBOA factors appear in the PMF analysis when 4 or 5 factor solutions are tested. In the 4-factor solution the COA was not present and the four factors were BBOA I, BBOA II, OOA, and HOA, Here, we present the 5-factor solution which also includes the COA factor. There are a few possible explanations about the two BBOA types. These factors may correspond to fresh and more processed BBOA or to different wood types or combustion conditions. This issue clearly deserves additional investigation. For the purposes of the current work though, the two BBOA factors could be even added to produce one total BBOA factor, since the reasons for the separation of the two by the PMF are not clear. The mass concentration of both BBOA I and BBOA II increased from 17:00 LT and remained at high levels until 1:00 LT (Fig. S5) which is consistent with the hours that fireplaces and wood stoves are used in the area. The highest BBOA concentration of the day was observed around 20:00 LT, when BBOA was almost 40 μg m$^{-3}$ on average.

The COA concentration peaked at the late evening hours (17:00- 23:00 LT) and had a weaker peak around midday (10:00-15:00 LT) (Fig. S5). The highest COA concentration was observed at 21:00 LT, which is the common wintertime dinner time in Greece, and was about 5 μg m$^{-3}$ on average. The COA spectrum was characterized by peaks at *m/z* values 41, 43, 55, 57 consistent with the results of Kaltsonoudis et al. (2017). The *m/z* 55 signal was almost twice that at *m/z* 57, a feature related to charbroiling in previous studies (Lanz et al., 2007; Allan et al., 2010).

The HOA concentration increased at 9:00 LT which is the local morning rush hour and during the evening at 18:00 LT. The HOA spectrum was characterized by the *m/z* values 41, 43, 55, 57, 67, 69 and 81 which are hydrocarbon fragments of typical traffic-related emissions (Aiken et al., 2009).

The OOA was characterized by a strong signal at *m/z* 44 (mostly due to $CO_2^+$) and another one at *m/z* 28, which is also reported in previous studies (Florou et al., 2017). The average diurnal pattern of OOA showed less variation compared to the primary factors. A slight increase of OOA was present during 10:00-14:00 LT, possibly due to the higher photochemical activity during that period.

### 4.1.2 Off-line

During the 2020 winter campaign in Patras, 31 ambient aerosol samples were collected on quartz filters using a medium-volume sampler (Tisch Scientific, 220 L min$^{-1}$ flowrate). The sampling started at 18:00 (LT) each day and lasted 24 h. The experimental procedure described in Section 2.1 was used for the off-line AMS analysis, and the off-line PMF was conducted according to the procedure described in Section 2.3.

The results of the PMF analysis for the off-line AMS method showed similar fractional contributions with the on-line (differences below 15%) (Fig. 4). The off-line BBOA was responsible for 48% of the total OA (Fig. 4). The second highest contributor to the total OA was COA (19%) and then HOA and OOA were responsible for 16% and 17% of the total OA respectively. These results are quite encouraging showing that our new off-line approach can reproduce the on-line results without the significant corrections (e.g. a factor of 10 for the HOA) needed in the original method.

The main difference of the above off-line analysis results with those of on-line is the different split of the two BBOA factors. The on-line PMF analysis indicated that BBOA I was 22% and BBOA II 31% of the total OA, while the off-line PMF analysis resulted in 32% for BBOA I and 16% for BBOA II (Fig. 4). However, the difference between the on-line and the off-line contribution to the total BBOA is below 5%, which shows that the off-line analysis can still provide an estimate of the importance of BBOA on the total OA that is consistent with the on-line results. This different split in the BBOA factors is probably of secondary importance given our limited understanding of the differences of BBOA I and BBOA II. For the purposes of the current work a difference of 15% is not that important and the sum of the BBOA factors could be viewed as the total BBOA.

The uncertainty of the off-line PMF results was estimated using bootstrap analysis. The results showed that BBOA I varied the most among the five factors. The estimated average contribution of BBOA I to the total OA was 32% and ranged between 18% (2.7 $\mu g\ m^{-3}$ based on the on-line OA concentration) and 47% (7 $\mu g\ m^{-3}$). All the other factor contributions to the total OA varied less than 20%. The BBOA II contribution to the total OA was 16% and varied between 11% (1.7 $\mu g\ m^{-3}$) and 21% (3 $\mu g\ m^{-3}$). The total BBOA estimated contribution to the total OA was 48% and ranged from 29% (4.4 $\mu g\ m^{-3}$) to 67% (10 $\mu g\ m^{-3}$). COA was on average 19% of the total OA and ranged between 11% (1.7 $\mu g\ m^{-3}$) and 28% (4.2 $\mu g\ m^{-3}$). The HOA contribution to the total OA was 16%, ranging from 12% (1.8 $\mu g\ m^{-3}$) to 21% (3.1 $\mu g\ m^{-3}$), and OOA was on average 17%, ranging from 11% (1.7 $\mu g\ m^{-3}$) to 23% (3.5 $\mu g\ m^{-3}$) (Fig. S6).

Off-line measurements of OC (e.g., by thermal optical analysis) in the same filter samples can be used to convert the fractions calculated above to absolute factor mass concentration in future studies. This standard analysis can be easily performed in parts of the collected filter samples. For example, the OC/EC analysis using a Sunset OC/EC analyzer requires just a small punch per filter (usually 1x1 $cm^2$ or 1x1.5 $cm^2$), so sample availability is not an issue. The organic aerosol (OA) concentration can then be calculated by multiplying the measured OC with the OA/OC that is determined from the high-resolution AMS measurements. The product of the factor fractions determined by our proposed method with the OA concentration will then give the mass concentrations of each factor.

Even though the average contribution of each factor to the total OA showed good agreement between the on-line and the off-line PMF results, higher discrepancies were observed on a daily basis (Fig. S7). For example, the off-line COA showed two different behaviors during the period, one of overestimation and one of underestimation compared to the on-line COA. This is evidence that the off-line AMS technique proposed in this work can give relatively accurate (within 15%) contributions of each factor for extended periods, but on a daily basis the uncertainty can be really high. This is due to a large extent to the temporal resolution of the off-line measurements (Vasilakopoulou et al., 2022). The reduced temporal resolution can result in significant source-apportionment differences for individual days. In Vasilakopoulou et al. (2022) discrepancies of a factor of 2 were observed for several factors on a daily basis. However, the source apportionment results were relatively accurate for a month-long period. We reach the same conclusion for the off-line measurements in this study. Another potential reason for these discrepancies is experimental issues such as the extraction efficiency of the filter samples, volatilization of semi-volatile organic compounds from the filter, etc.

Even though the factor contributions of the off-line and the on-line PMF results differed by less than 15%, the factor profiles were significantly different. The off-line PMF spectra were compared with the on-line, using the theta angle approach (Kostenidou et al., 2009). Theta is the angle that the two factors have if they are treated as vectors. The highest difference was observed for HOA ($\theta=40°$) (Fig. S8), showing that the off-line HOA spectrum is quite different from the on-line. This is not surprising given the limited water solubility of the HOA components. However, the HOA tracers ($m/z$ values 41, 43, 55, 57

and 69) were present in the off-line HOA spectrum making its identification possible. These differences are probably due to the partial inclusion of the HOA components in the aerosol that are detectable by the AMS. Therefore, the extraction process appears to modify the AMS spectrum, but still the PMF reaches a reasonably accurate estimate, at least in this case, with the "processed" AMS spectrum. Also, high theta angles were observed for the other primary factors ($\theta>25°$). The off-line COA spectrum differed by 34° from the on-line one, the off-line BBOA I by 27°, and the off-line BBOA II by 36° from the on-line

BBOA II. The OOA had the most similar spectrum profile between the off-line and the on-line results ($\theta=17°$). A possible explanation about this similarity is that secondary OA is much more soluble in water than COA and HOA.

        These differences in the spectra profiles are a consequence of two facts. First, the lower temporal resolution that the off-line results have (24 h) compared to the on-line (3 min) can affect the profiles of each factor leading to high theta angles that can reach up to 30° (Vasilakopoulou et al., 2022). The reduction of the information provided to the PMF moving from

high temporal resolution (usually thousands of measurements) to daily resolution (hundreds or even less) measurements affects the results. Also, when daily averages are used the diurnal variation of each factor is lost. Second, the extraction efficiency of the different OA components can also result in increased angles.

        In order to understand better the differences between the off-line and the on-line PMF analyses, we compared the total organic mass spectra from the two techniques. The off-line organic mass spectra were blank corrected. The theta angle

between the average off-line and the average on-line organic mass spectrum was 21° (Fig. 5). Significant differences were observed at $m/z$ values below 20, since the theta angle drops to 17° if these values are not accounted. These $m/z$ values should be checked carefully because they may affect the results. Water appears to contribute to these differences in the low $m/z$ values. For example, the $m/z$ 18 is higher in all off-line spectra compared to the on-line. The exclusion of these values from the PMF factors reduced the theta angle by 3°.

Comparing the signals at four major HOA fragments ($m/z$ 55, 57, 67, 69) we estimated that on average the off-line technique captured 64% of the $C_xH_y^+$ and 82% of the $C_xH_yO^+$ (Fig. 6). The average off-line to on-line ratio for these HOA markers was 0.73 (Fig. S9). The $C_xH_yO^+$ fragments contain oxygen, so they should be coming from relatively water-soluble compounds. The difference in the two approaches is 18%, suggesting that indeed the off-line AMS approach captures these compounds to a very large extent. The $C_xH_y^+$ fragments are characteristic of hydrocarbon-like OA (HOA) that has in general

low water solubility. The fact that 64% of the material is present in the off-line organic mass spectra provides strong evidence that our approach extracts and includes in the corresponding AMS analysis the majority of these mostly water insoluble compounds. If these compounds remained on the filter there would be no signal at the corresponding $m/z$ values for $C_xH_y^+$.

A day-by-day analysis has also been performed between the on-line (24 h averages) and the off-line samples. The organic mass spectra comparison between the off-line and the on-line results showed that the best agreement was observed on January 20 and their angle was 16°. The worst agreement was observed on January 30 and the theta angle was 26° (Fig. S10).

The atomic oxygen to carbon ratio (O:C) of the off-line spectra was on average 0.47 and the on-line was 0.50, showing good agreement between the two for the whole period (Fig. S11). The fractional error of the off-line compared to the on-line O:C was 12%.

The organics to sulfate ratio (OA/Sulf) was also compared between the on-line and the off-line results. Organics in general, have a wide range of water solubility. They can be either soluble, insoluble or partially soluble in water. Sulfate on the other hand is completely soluble in water for the conditions of the extraction. So, comparing the two can give us an estimate of the fraction of the organics that was analysed with the off-line AMS technique. The off-line (OA/Sulf) ratio followed the same trend over time of the on-line ratio (Fig. 7). The two ratios agreed within 30%. The off-line results showed a surprisingly small overestimation of the OA/Sulf ratio compared to the on-line with a bias of 1.05 and a fractional bias of 0.18. The average off-line OA/Sulf ratio was 8.0, while the on-line was 7.0.

## 4.2 Source apportionment of the Patras 2019 summer campaign

### 4.2.1 On-line

Solutions with one up to four factors were examined and the three-factor solution was chosen as the one which explains better the ambient OA spectral variation. One primary (HOA) and two secondary (MO-OOA, LO-OOA) factors were identified (Fig. S13). The OA was highly oxidized during the examined period, as the secondary factors accounted for 80% of the total OA (LO-OOA 48% and MO-OOA 32%). The estimated average HOA contribution to the total OA was 20% (Fig. 2).

The HOA spectrum was characterized by *m/z's* 41, 43, 55, 57, 67, 69 and 81 (Fig. S14). Its diurnal profile had one peak at 10:00 LT and one during the evening at 22:00 LT (Fig. S15). The HOA O:C was 0.22, which is on the high end of the ranges reported in the literature. Since HOA and COA have relatively similar AMS spectra, distinguishing between the two is not always possible. The HOA of this study agrees well with the HOA-2 reported in Kostenidou et al. (2015), which also included meat cooking emissions.

MO-OOA was the most oxidized of the two secondary factors. Its diurnal profile had small variation, showing that it was not affected by local sources. On the other hand, the LO-OOA diurnal profile was more variable and was characterized by a first peak at 10:00 LT and a second one during the evening (22:00 LT). Both the MO-OOA and LO-OOA mass spectra were characterized by a strong peak at *m/z* 44. However, the LO-OOA spectrum had also peaks at *m/z* values, 39, 41 and 55. These peaks were weak in the MO-OOA spectrum. Also, the MO-OOA O:C was a little higher (1.0) than the O:C of the LO-OOA (0.94). Their angle was 9°.

## 4.2.2 Off-line

We focus here on the analysis of the 31 daily samples collected during the early summer period (March to June 2020) in Patras. A low-volume sampler (6.7 L min$^{-1}$) was used for the sample collection on quartz filters and the sampling period was 24 hours. Because of the low mass loading of the filter half of the initial filter (47 mm) was used in this off-line AMS analysis.

The off-line PMF analysis resulted in a three-factor solution with one primary (HOA) and two secondary (MO-OOA and LO-OOA) factors. The off-line results showed that the OA was highly oxidized, which is in agreement with the on-line results. OOA was responsible for 68% of the total OA, while HOA represented 32% of the total OA, which is 12% more than in the on-line solution. The contribution of each factor to the total OA differed by less than 17% with the on-line estimation (Fig. 8). The off-line HOA contribution to the total OA was 32% and the bootstrap analysis showed that it ranged from 23% to 42% (5th and 95th percentiles). The MO-OOA was 37%, ranging from 24% to 57%, and the LO-OOA was 31% (ranging from 26% to 35%) (Fig. S16). These uncertainties suggest that the estimates of the two approaches were practically in agreement with each other considering their uncertainty.

The off-line PMF spectra were characterized by the specific markers of each factor. The off-line HOA spectrum was characterized by peaks related to aliphatic hydrocarbons i.e., 41, 43, 55, 57, 67 and 69 (Fig. S17). The MO-OOA spectrum was characterized by the $m/z$ 44 peak, and the LO-OOA by the $m/z$ peaks at 43 and 44. The different $m/z$ 43/44 ratio among the two secondary factors was used for their identification. The off-line spectrum of each factor was different compared with the on-line spectrum. The off-line HOA spectrum differed by 34° from the on-line HOA (Fig. S18). A high theta angle was also observed for the LO-OOA, as the off-line LO-OOA differed by 34° with the on-line LO-OOA. The MO-OOA spectra were more similar, as the off-line one differed by 12° with the on-line MO-OOA spectrum. The O:C of the MO-OOA was 1.2 and of the LO-OOA 0.81 in this case, and their angle was 35°. These changes in the spectra of the factors appear to be characteristic of the low-temporal resolution off-line analysis. The differences of the LO-OOA and MO-OOA spectra vary widely in different campaigns because they appear to represent roughly the upper and lower limits of SOA oxidation encountered in the specific campaign.

The average total off-line to on-line spectra comparisons in the summer period, showed better agreement compared with the winter campaign. We believe that this behaviour is partially due to the mixing state of the particles (e.g., the co-existence of water-soluble secondary PM components with the water-insoluble primary combustion material). The different mixing state in each period is an important factor affecting the off-line measurements. For example, during the summer the primary particles from transportation are rapidly covered with sulfates and secondary organic aerosol. This process is slower during the winter, so there are particles that do not include much water-soluble material. Our hypothesis is that this water-soluble material when present facilitates the transfer of the water insoluble particle core from the filter to the water extract. The total OA spectra for the off-line and the on-line results had an average theta angle of 12° (Fig. S19). The best agreement was observed on May 8, when the two spectra differed by 7°. The worst was observed on May 18 and the angle between the off-line and the on-line OA spectra was 16°.

The off-line technique captured a big part of the HOA on-line signal as indicated by the comparison of five HOA markers (Fig. 9). For these HOA markers the average $C_xH_yO^+$ off-line to on-line ratio was 0.77, and the average $C_xH_y^+$ off-line to on-line ratio was 1.1 (Fig. 9). This good agreement between the off-line and the on-line HOA markers could be responsible for the lower theta angle between the off-line and the on-line HOA spectrum in the PMF analysis, compared to the winter campaign.

### 4.3 Source apportionment of the Athens 2019 winter campaign

#### 4.3.1 On-line

Five factors were identified, three primary (HOA, COA, BBOA) and two secondary (MO-OOA, LO-OOA). The primary factors represented on average 53% of the total OA, and the secondary 47%. The contribution of the secondary factors to the total OA was relatively high for a winter period in the center of a big city. The bootstrap analysis suggested low uncertainty for the contribution of each factor to the total OA (below 15%) (Fig. S21).

The BBOA (11% of the total OA) had high concentrations during January and February and decreased in March (Figs. S22-S23), a behavior that is consistent with the temperatures in Greece during the winter period. The highest hourly BBOA mass concentration (26 µg m$^{-3}$) was observed on February 19 at midnight, which was also the night with the highest OA mass concentration. The same night the HOA mass concentration was the highest of the three-month period (49 µg m$^{-3}$). On average the HOA mass concentration was 2.1 µg m$^{-3}$ and it represented 26% of the total OA. On the other hand, COA (16% of the total OA) remained at the same levels during the three-month period examined. The LO-OOA (21% of the total OA) and the MO-OOA (26% of the total OA) were less variant during the three-month period compared to the primary factors.

#### 4.3.2 Off-line

In this campaign 33 daily samples were collected in Thissio, at the National Observatory of Athens. A high-volume sampler (DH-77, Digitel) was used for the sample collection. One of the additional objectives of this test was to study the difference that may occur between HR-AMS off-line measurements and unit mass resolution ACSM online results.

The average off-line mass spectrum was similar to the on-line ($\theta=16°$) (Fig. S24) for the examined days. The off-line PMF solution resulted in five factors, just like the on-line solution. Three primary (HOA, COA and BBOA) and two secondary (MO-OOA, LO-OOA) factors were identified. Each factor was identified using the specific tracer *m/z's* that have been already discussed in the previous sections. The off-line spectrum of each factor was different compared to the one in the on-line PMF solution (Fig. S25). The difference in the estimated spectra are partially due to the different sampling temporal resolution (30 min in the on-line, daily in the off-line) (Vasilakopoulou et al., 2022) and could also be partially due to the different instrumentation used for the on-line and the off-line analysis. In the off-line PMF results of this study, the two secondary factors start to mix at low temporal resolution PMF results, as the O:C of the off-line MO-OOA is lower compared to the on-line, while the O:C of the LO-OOA is higher. This is also characteristic of the reduced temporal resolution analysis

(Vasilakopoulou et al., 2022). In order to estimate the O:C from the on-line ACSM measurements we used the approach of Canagaratna et al. (2015) for the unit-mass resolution data. The O:C estimates are quite uncertain in this case. The O:C of the LO-OOA was 0.7 and the O:C of the MO-OOA was 0.87 in the off-line analysis. By definition LO-OOA always has a lower O:C compared to that of MO-OOA. The LO-OOA and MO-OOA separation is performed by the PMF and they can always be summed to obtain the total OOA. We present and discuss both of them in the present study to facilitate comparisons with the result of the online analyses in the literature.

The difference between the on-line and the off-line contributions to the total OA were below 15% (Fig. 10), which is evidence of consistency between the off-line HR-AMS analysis with the on-line ACSM results. The highest difference was observed for BBOA, which was 13% higher (24% vs 11%) than the on-line BBOA, and the lowest was observed for COA, which was 1% lower than the on-line COA (15% vs 16%). The off-line HOA was 8% lower (18% vs 26%). The off-line LO-OOA contribution to the total OA was only 4% higher (25% vs 21%) that the on-line LO-OOA. The off-line MO-OOA contribution to the total OA differed by 9% (17% vs 26%) from the on-line MO-OOA. These differences are inside the uncertainty range of the off-line analysis.

In order to estimate the uncertainty of the off-line PMF results a bootstrap analysis of 1000 runs has been performed. The mean BBOA contribution to the total OA was 24% and ranged from 14% to 36% (Fig. S26). The HOA contribution to the total OA ranged from 14% to 23% (the mean value was 18%). The estimated average COA contribution to the total OA was 15%, and it ranged from 9% to 21%. The MO-OOA ranged from 15% to 20% (mean value 18%) and LO-OOA ranged from 16% to 37% (mean value 25%).

A sensitivity analysis has also been performed, in which the off-line high-resolution AMS spectra were averaged to unit resolution and were used as inputs to the PMF. The UMR PMF analysis resulted in more mixed profiles than the HR analysis and the factors were not identified as easily as in the HR analysis. The contribution of each factor to the total OA between the HR and the UMR off-line PMF results differed by less than 20% (Suppl. Mat. Section S4). The highest difference was observed for BBOA and was 17% (24% for the HR analysis and 7% for the UMR analysis). Comparing the UMR off-line solution with the on-line solution the highest difference observed was 14% and it referred to the COA. The UMR off-line COA was 21% of the total OA, while the on-line COA was 17%.

## 5 Transfer of OA from the filters to the AMS

Our results so far have indicated good agreement between the off-line and collocated on-line AMS measurements. This is a rather surprising result, because a significant fraction of the organics analysed (components of cooking aerosol or components of the HOA coming from transportation) are practically insoluble in water. The coating of these particles with soluble secondary material can probably explain the corresponding results during the different campaigns. Our hypothesis is that during the extraction process, the water-soluble material of the particles dissolves, the remaining small water-insoluble particle cores leave the filter and get suspended in the water during the sonication phase and then are transferred with the solution to

the atomizer. After they enter the atomizer, they are included in the formed water droplets together with the dissolved material and become part of the produced aerosol after the drying. In this way they are measured by the AMS. The different aspects of this hypothesis are tested in the following sections.

### 5.1 Measurements of suspended particles in the liquid phase

In order to test our hypothesis, we first quantified the concentration of the particles in the aqueous phase after the extraction and the filtration phase. Filtration (1 μm pore size) was used only for the measurements of the suspended particles in water and not for the off-line AMS analysis. If our hypothesis is not valid, there should be no particles suspended in the solution, just the background. We used the Zetasizer to measure the size distribution of nanoparticles in the water using dynamic light scattering. The particle size distribution was measured in the ultrapure water used for the extraction and in the solution produced by the extraction of ambient aerosol samples. The Zetasizer was calibrated using monodisperse polystyrene spheres (PSL) of 100 nm and 200 nm (Suppl. Mat. Section S5).

The method was applied to 20 ambient samples, which were collected during winter at the University of Peloponnese campus, in Patra. The results support our hypothesis as significant concentrations of particles of sizes in the 100-200 nm size range were detected in the extract of the ambient aerosol samples (Fig. 11). These particles were not present in the blanks. In this case only the water was used as blank.

### 5.2 Insoluble material removed from the filter after the extraction process

A significant number of suspended particles was measured in the water extracts using the Zetasizer. However, with that test we could not get information about the chemical composition of the particles leaving the filter. In this section we focus on the least water-soluble component of the particles: elemental carbon. Since the Zetasizer results showed that particles were detected in the water extracts, the concentration of insoluble material on the filter after the extraction process, should be reduced compared to the initial concentration.

To test this, we used a Sunset OC/EC analyzer for 20 ambient filter samples. In these additional experiments the EC was measured before and after the extraction. The filters after the extraction were dried at 100°C for 5 min before the EC measurement, because the humidity of the sample increases significantly the uncertainty of the EC measurement. The ambient EC concentration for the examined period was on average 0.7 μg m$^{-3}$ and ranged from 0.2 to 1.4 μg m$^{-3}$. A significant fraction of EC was missing from the filter after the extraction process. The average EC on the filter after the extraction was 44% of the initial (Fig. 12). The significant reduction of the EC on the filter supports our hypothesis that the insoluble particle cores leave the filter and get suspended in the water.

### 5.3 BC measurements using the off-line technique

To test if the insoluble material can also survive through the atomization and drying processes, we measured the BC in the resulting particles with a Single Particle Soot Photometer-Extended Range (SP2-XR) (Droplet Measurement Technologies). We focus on BC because it is water insoluble and easy to measure and thus can be used to prove that the suspended insoluble material can be transferred to the AMS. Two filter punches (3 cm$^2$ in total) from ambient samples were extracted in 10 mL of ultrapure water, following the off-line procedure. The samples were placed in an ultrasonic sonicator for 30 min and no filtration phase was performed. The water extracts were then atomized and dried, as in the off-line AMS procedure, and then the resulting aerosol was detected with the SP2-XR. Significant BC concentrations were detected in the samples (Fig. 13), while in the blank (atomized water) the BC was practically zero. These tests strongly support our hypothesis that a significant fraction of even the most insoluble material (elemental carbon) in the ambient particles, leaves the filter, gets suspended in the aqueous solution as particles in the 50-300 nm size range and makes it to the AMS.

### 6 Conclusions

The off-line AMS technique can be a powerful tool for characterizing OA in areas and periods when an AMS is not available. However, so far it could capture only the water-soluble part of OA if no corrections were used. This could lead to significant differences of the source apportionment results compared to on-line measurements. In this work an improved off-line aerosol mass spectrometer analysis technique has been developed, which can capture a significant part of the insoluble and the partially soluble fraction of OA.

The improved off-line AMS technique has been evaluated in three different campaigns in Greece, two during winter (Athens and Patras) and one in the summer (Patras). PMF analysis was performed for each campaign, separately for the on-line and for the off-line results. The fractional contribution of each source to the total OA differed by less than 15% between the on-line and the off-line PMF results, which shows that the two methods are in good agreement (considering their uncertainty) without any corrections required for the off-line results.

The AMS spectra, however, showed significant differences between the on-line and the off-line PMF results. This is due to both the reduced temporal resolution in the off-line results compared to the on-line, and to the uncertainty introduced by aspects of the experimental procedure, such as the extraction efficiency etc. However, the PMF appears to be able to adjust the factors and still give relatively accurate fractional contributions of each factor to the total OA. Comparing summer with winter results, better agreement was observed in summer between the on-line and off-line OA spectra. This could be due to the prevalence of the secondary material that coats most of the primary particles and potentially facilitates their suspension as it dissolves in water.

The proposed improved off-line AMS technique can mobilize a significant fraction of the water insoluble material and allow the AMS to measure it. This reduces significantly the uncertainty compared to the original method. The suspension of the insoluble material in the improved off-line technique, was evident by the considerable number of particles detected in

the water extract at the range of 50-300 nm, and by the lack of more than half of the EC from the filter after our water extraction process. In addition, a significant BC concentration was detected in the aerosol samples before the AMS, which is evidence that insoluble material can be the extracted from the filter, can be atomized and can get analysed by the AMS with the proposed approach.

*Data availability.* Measurement data are available by request by ch.vasilakopoulou@chemeng.upatras.gr.

*Author contributions.* CNV and SP designed the study and wrote the paper. CNV did the off-line measurements and analysis. CNV, KF, CK, IS and NM obtained and provided the on-line measurements. All authors edited the manuscript.

*Acknowledgments.* We would like to thank Dr. Aikaterini Bougiatioti and Dr. Despina Paraskevopoulou for providing the filters from the Thissio 2019 winter campaign.

*Financial support.* This research has been supported by the EU H2020 RI-URBANS project (grant 101036245) and by the Hellenic Foundation for Research & Innovation (HFRI) under project CHEVOPIN, grant agreement no. 1819.

*Competing interests.* None.

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

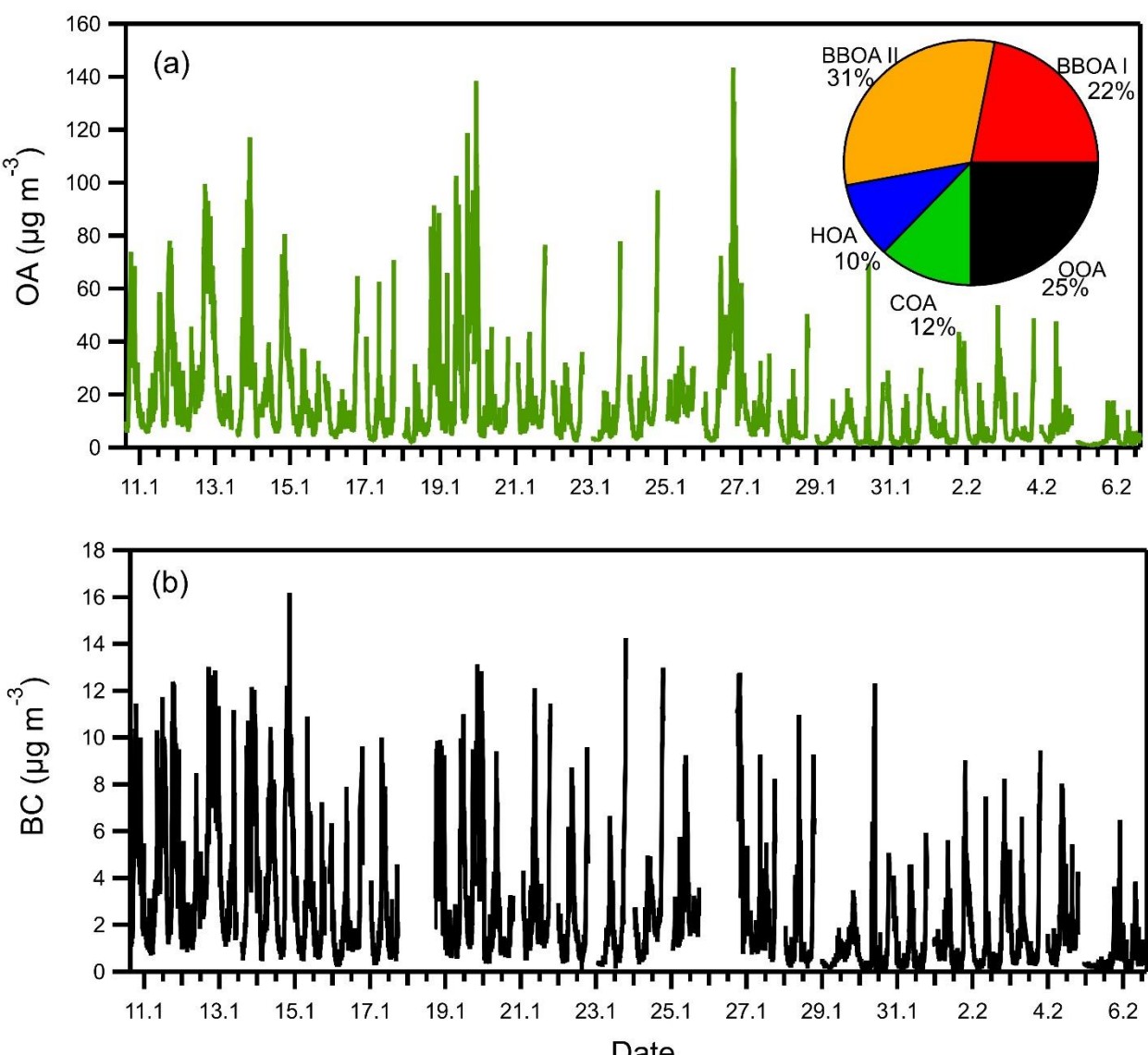

**Figure 1:** Time series of a) OA and b) BC during the winter 2020 campaign in Patras. The fractional contribution of each factor to the total OA is also shown.

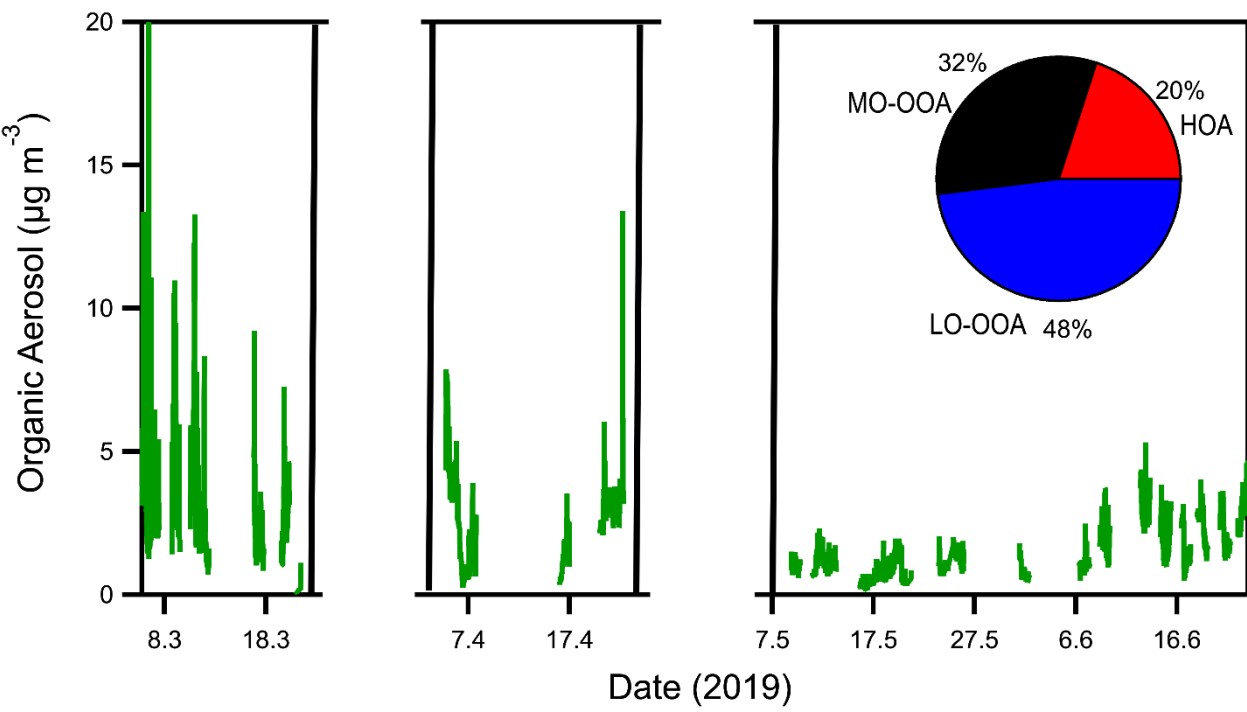

**Figure 2:** Organic aerosol time series and fractional composition of the sources derived from the PMF analysis of the on-line measurements for the summer 2019 Patras campaign. Measurements were performed only during specific days in this three-month period.

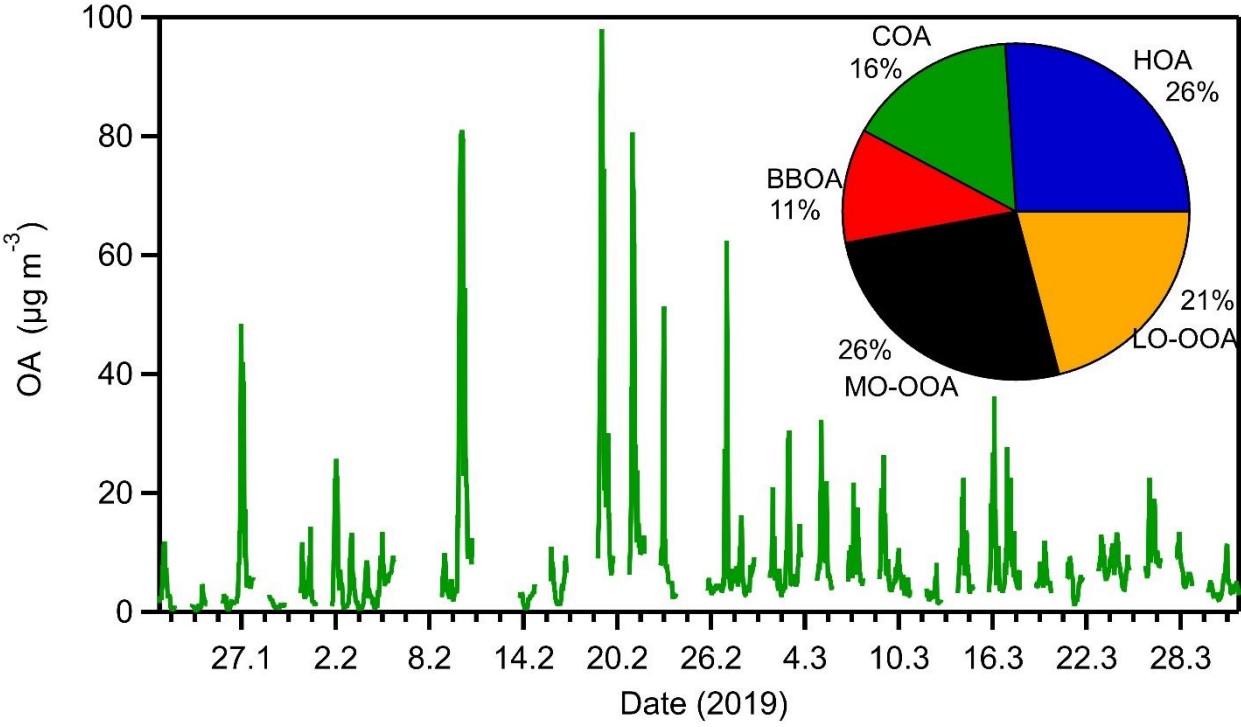

**Figure 3:** Organic aerosol concentration and fractional composition of the sources derived from the PMF analysis for the on-line measurements for Athens 2019 winter campaign.

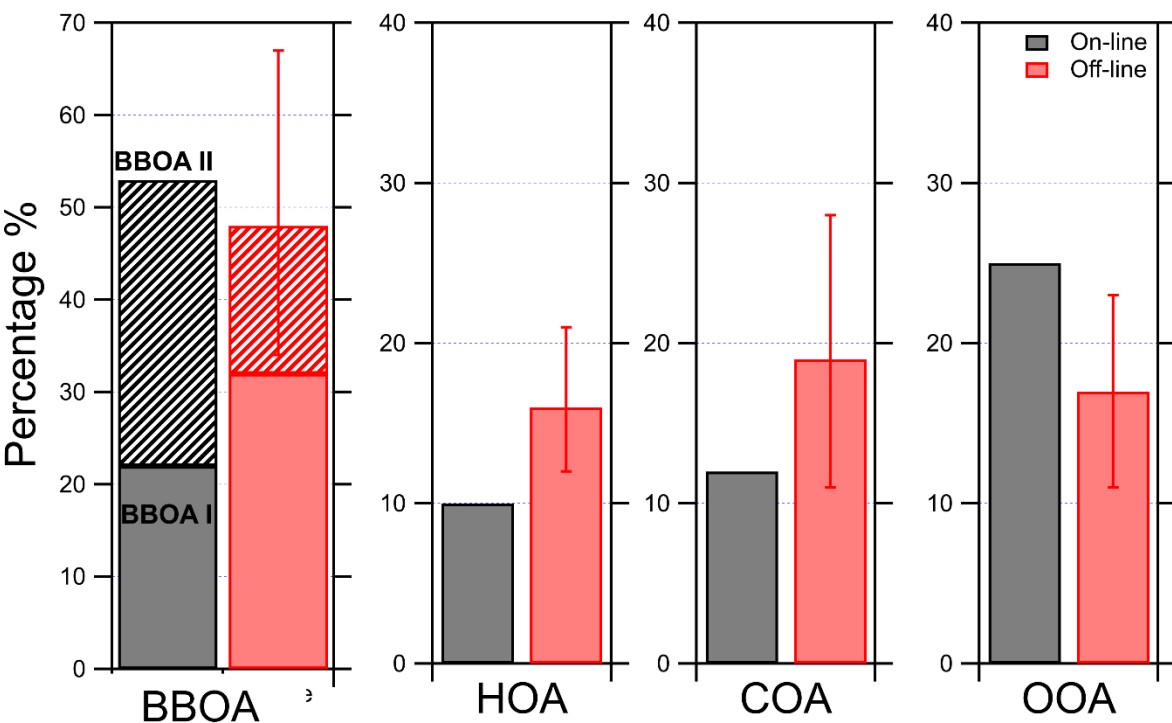

**Figure 4:** Contribution of each factor to the total OA for the on-line and the off-line PMF results in Patras during the winter of 2020. The contribution of each one of the two BBOA factors to the total OA is also shown. A different scale is used for the BBOA.



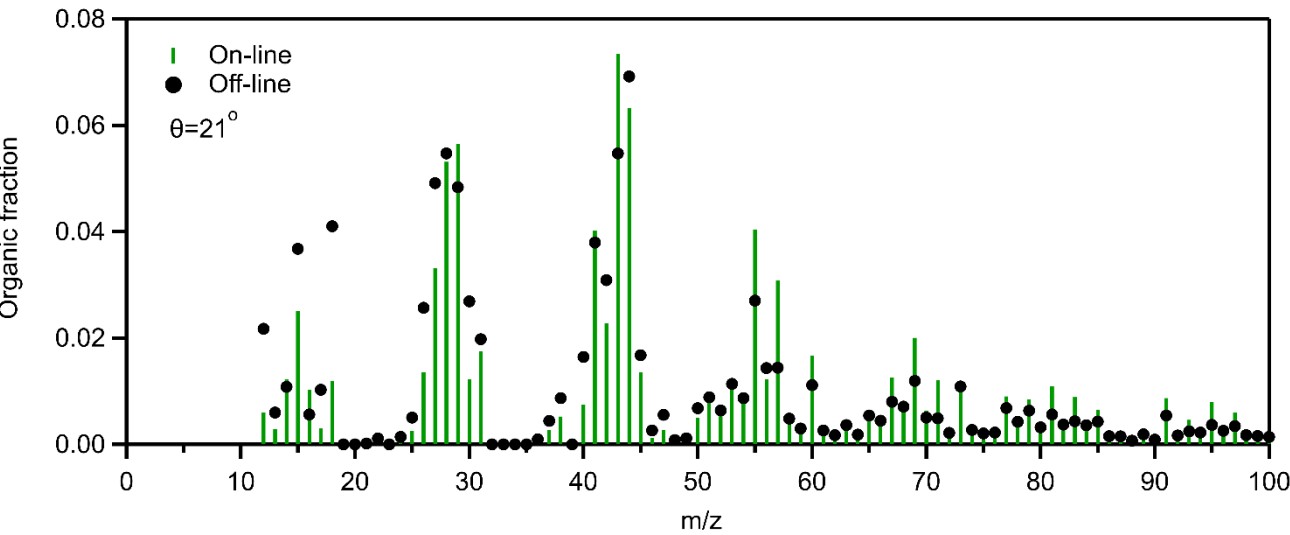

**Figure 5:** Comparison of the average on-line and off-line organic mass spectrum for the Patras 2020 winter campaign.



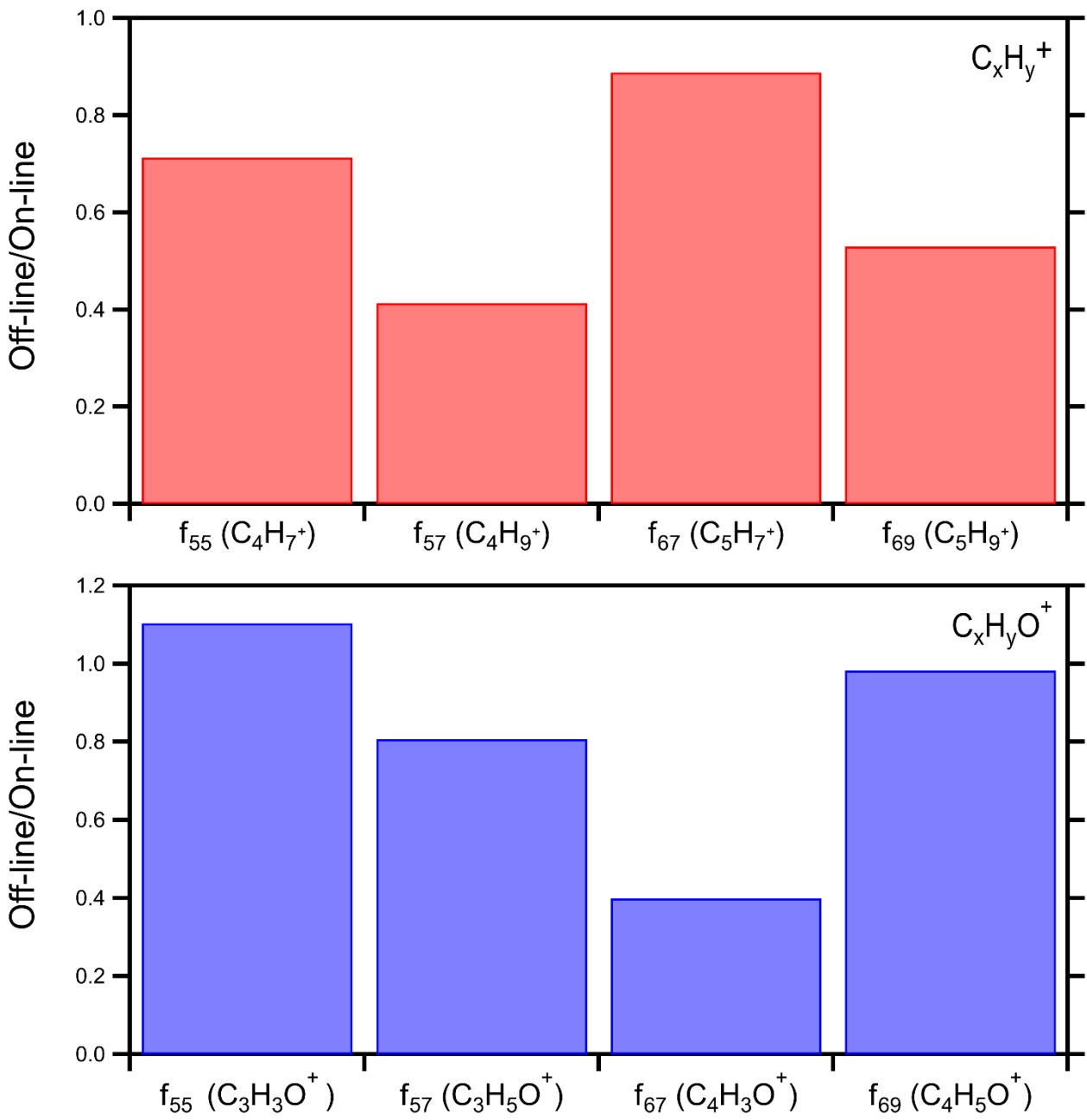

**Figure 6:** Off-line to on-line ratio of various HOA markers ($C_xH_y^+$ on top and $C_xH_yO^+$ at the bottom) for the winter 2020 Patras campaign.

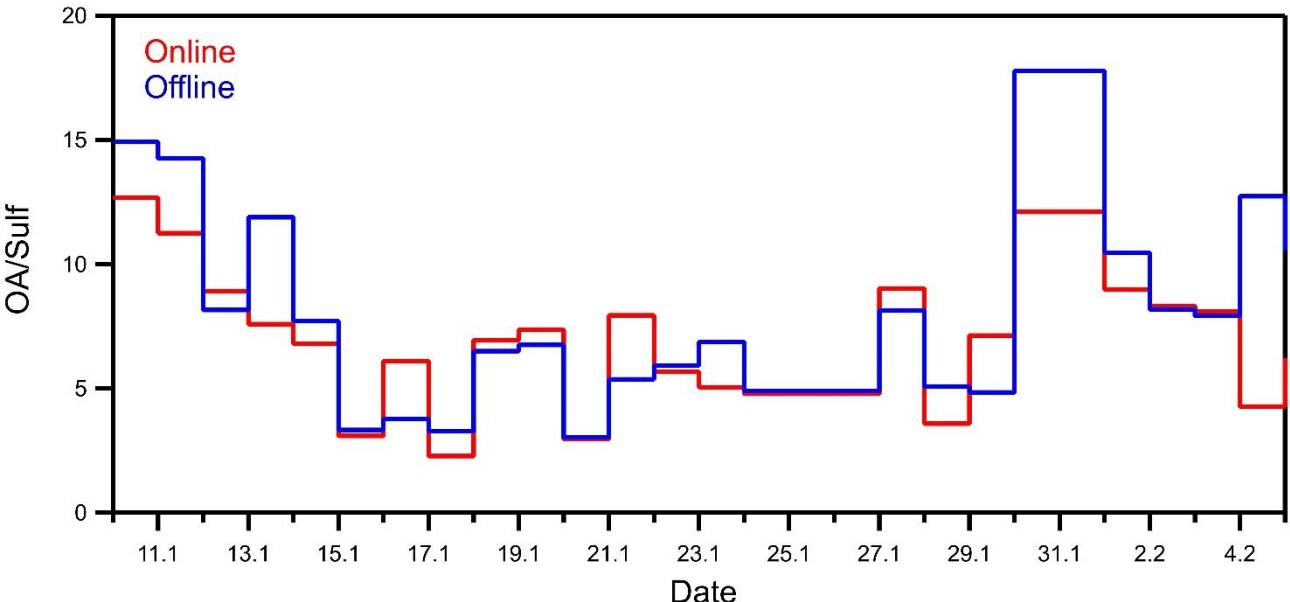

**Figure 7:** Organics to Sulfate ratio comparison between the off-line and the on-line results for the Patras winter 2020 campaign.



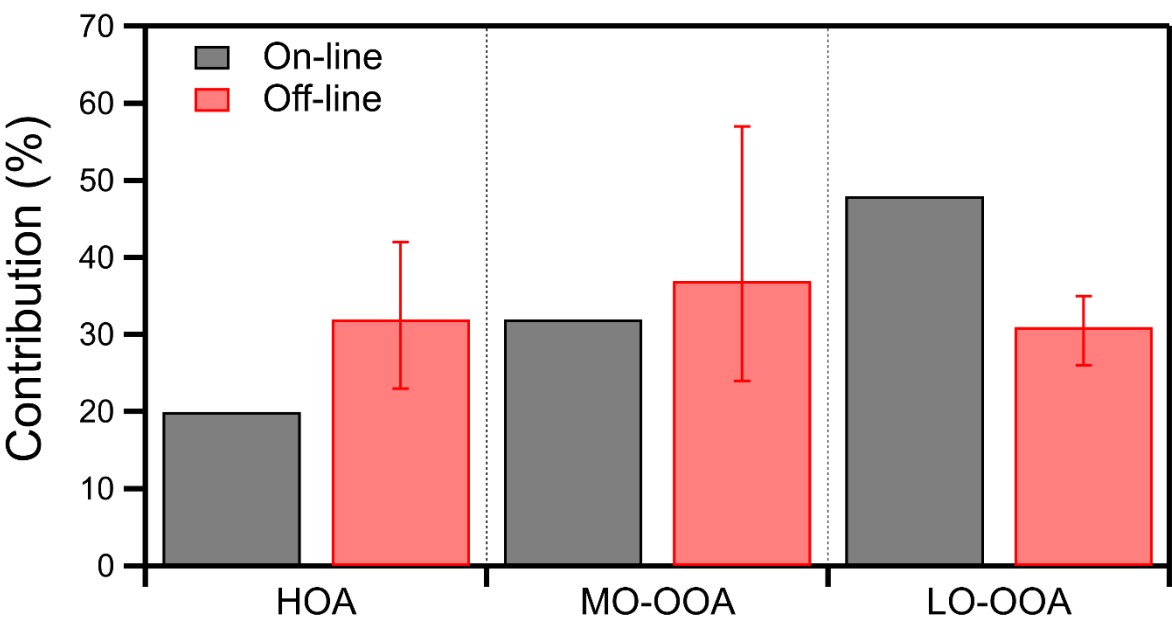

**Figure 8:** Comparison of the contribution of each factor to the total OA between the on-line and the off-line PMF results for the summer 2019 Patras campaign.




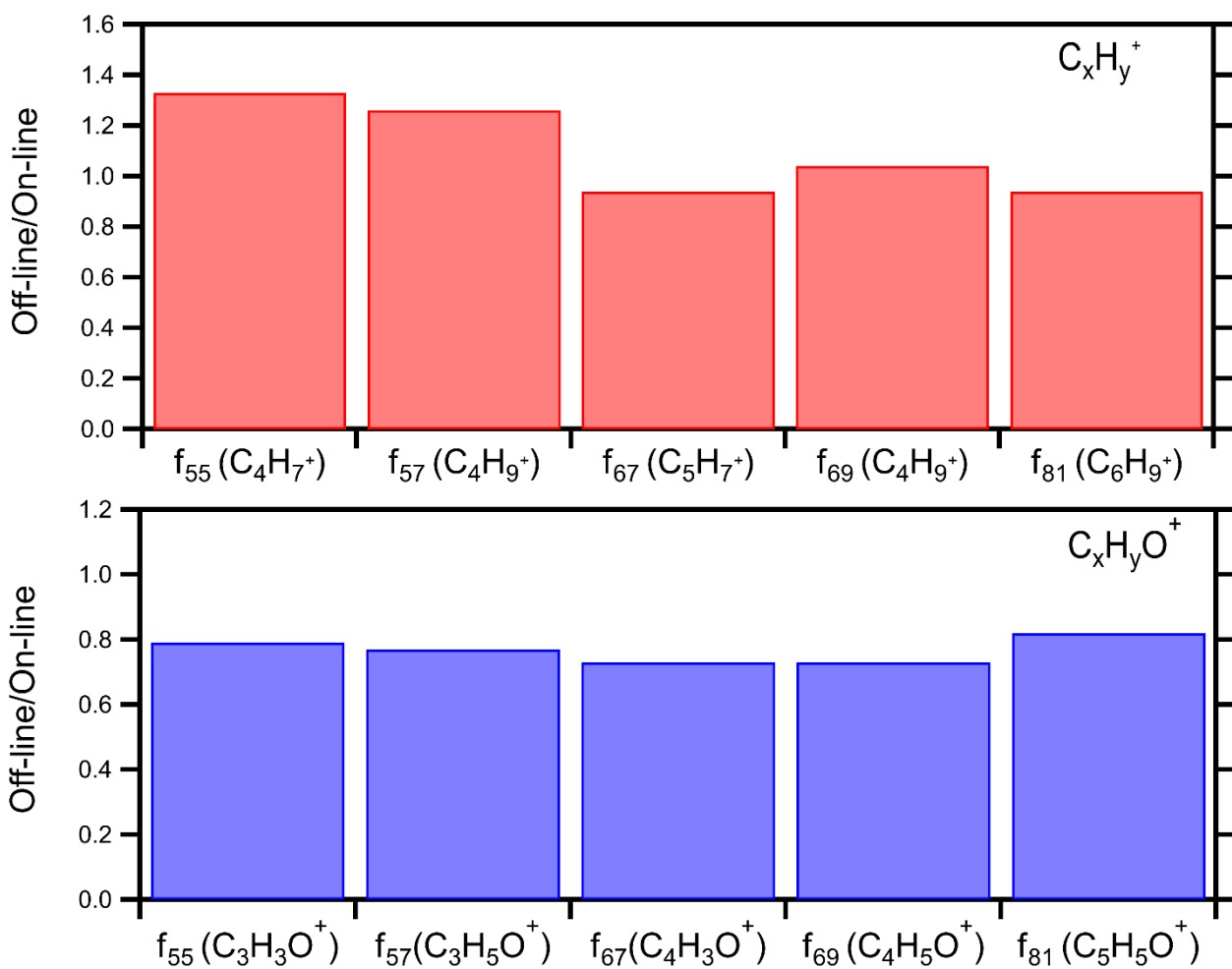

**Figure 9***: Off-line to on-line fraction various $C_xH_y^+$ and $C_xH_yO^+$ HOA markers for the summer 2019 Patras campaign.*




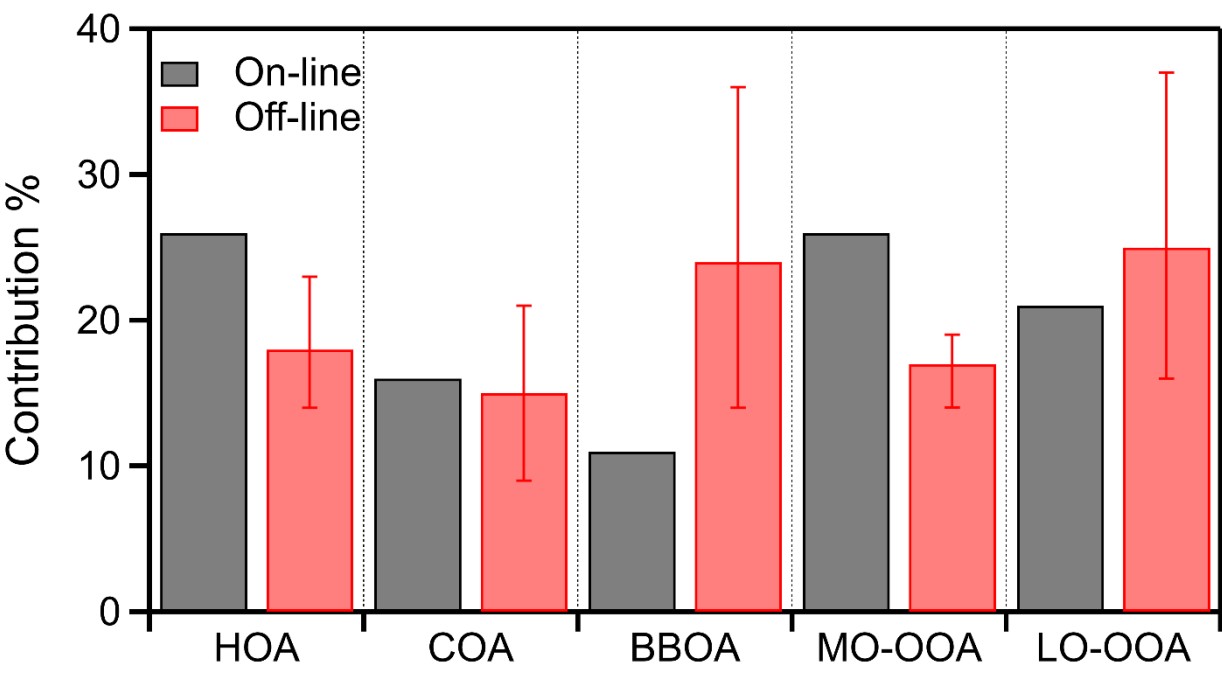

**Figure 10:** Comparison of the contribution of each factor to the total OA between the on-line and the off-line PMF analysis for the Athens 2019 winter campaign.



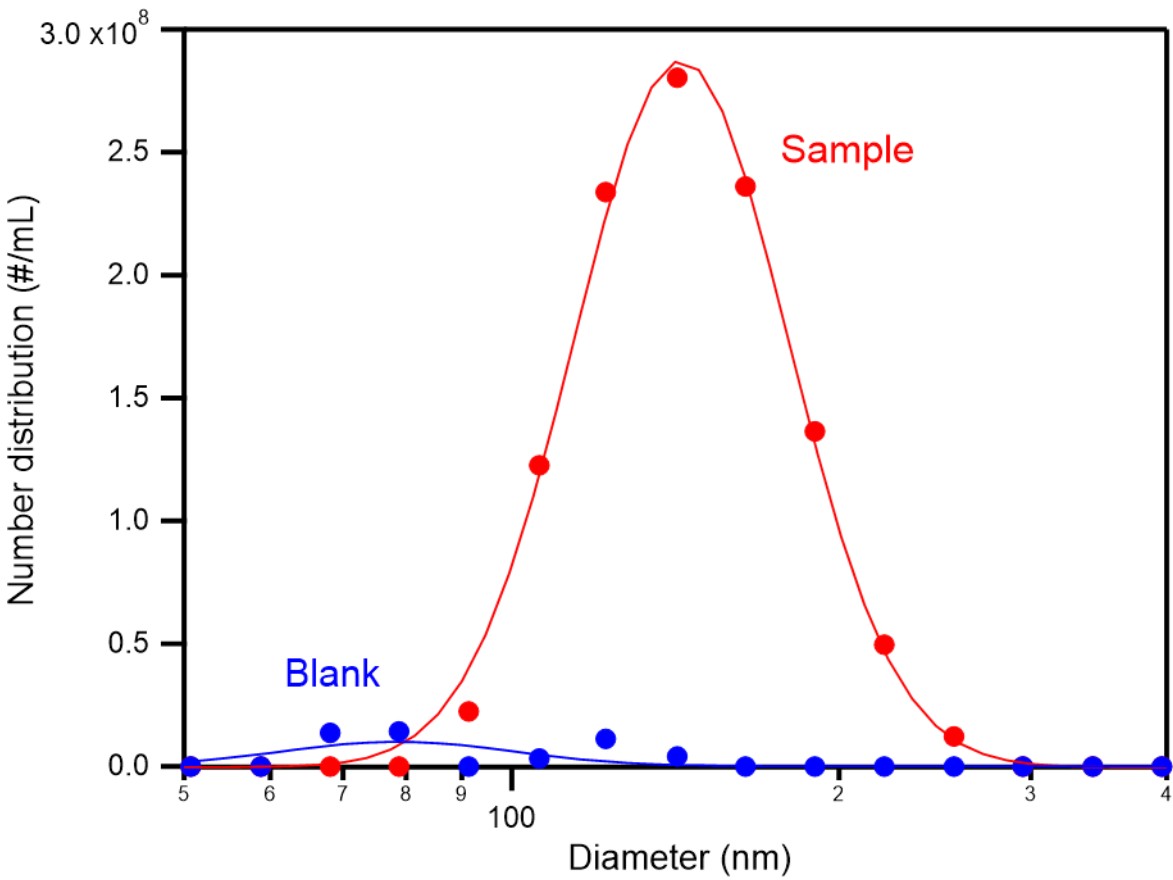


**Figure 11:** Number concentration of suspended particles in the water extract for the sample (in red) and the blank (in blue).



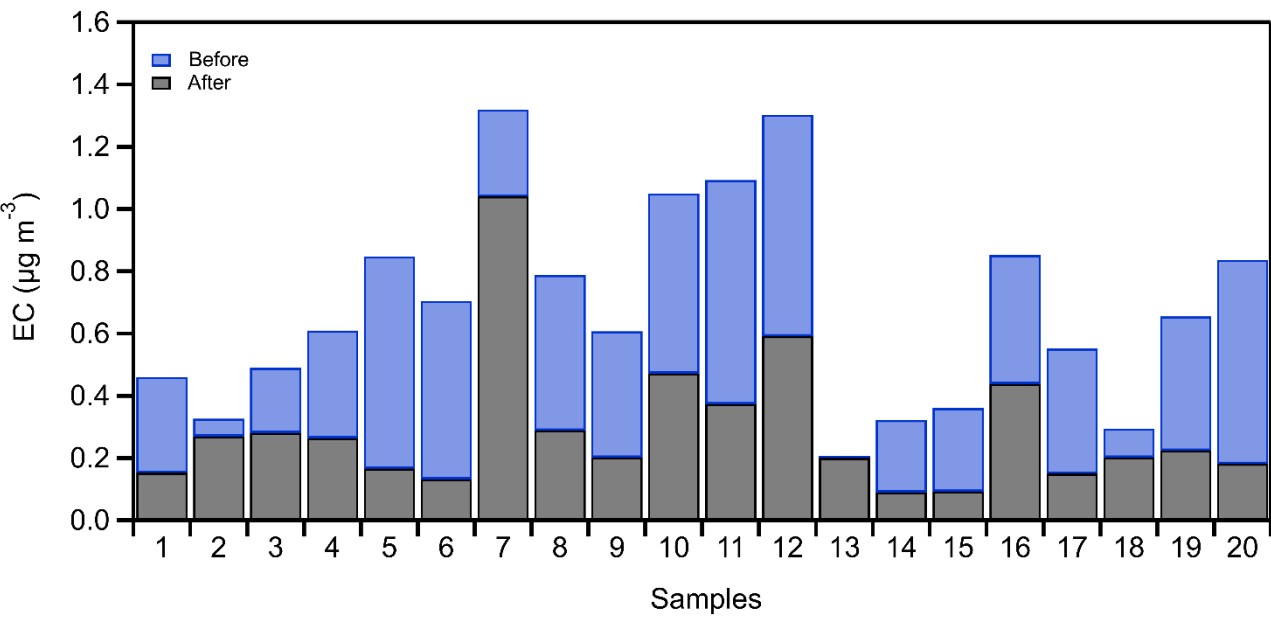

**Figure 12:** Elemental carbon before and after the extraction process.




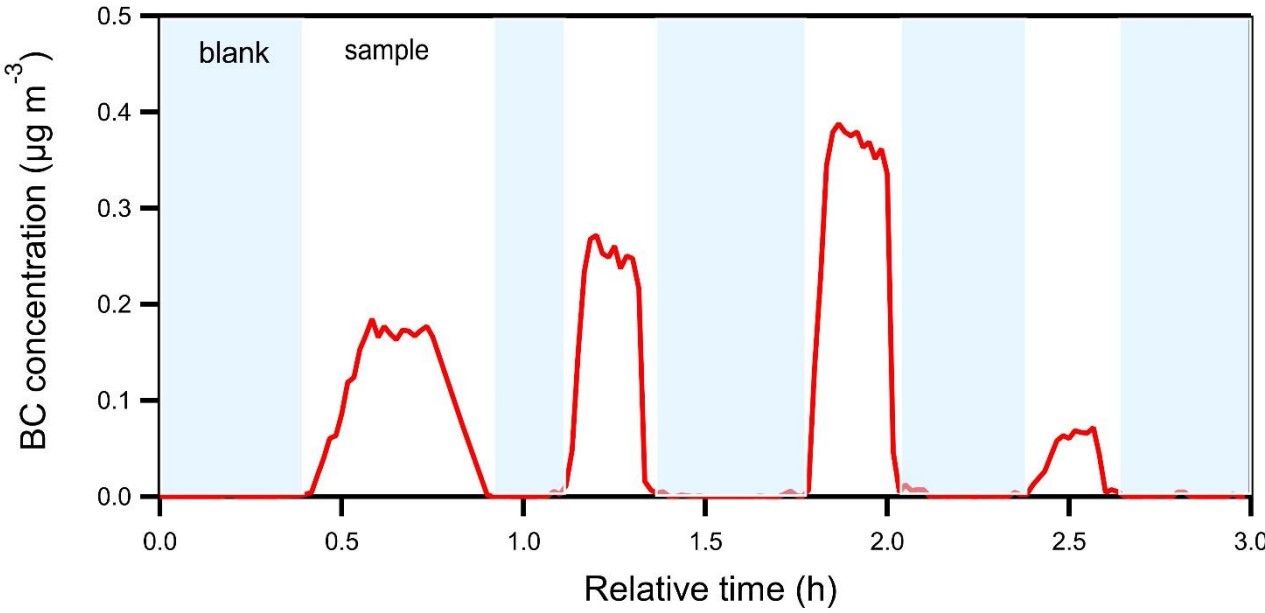


**Figure 13:** BC concentration time series of the off-line SP2 test. The blanks are shown in blue background and the samples in white.