# Peer review of "Development and Evaluation of an Improved Off-Line Aerosol Mass Spectrometry Technique"

_Atmospheric Measurement Techniques, 2023_

## Author Response (AR1)

**Responses to the comments of the Reviewers**

**Reviewer 1**

**(1)** This manuscript describes a method modification to off-line extraction of filter-based particle collections with re-suspension and analysis by high resolution AMS. Importantly, the method has been compared to on-line AMS (and ACSM) measurements from 3 different field studies. One limitation of the work presented here is that the off-line analysis is non-quantitative under the current method. However, the relative amount of contributing PMF components/factors shows reasonable agreement between the on-line analysis and off-line analysis.

We appreciate the helpful suggestions and comments from the reviewer. Our responses (in black) and the corresponding changes in the manuscript follow each comment of the reviewer (in blue).

Our proposed off-line method can easily become quantitative, if the corresponding samples are also analyzed for OC/EC (e.g., by thermal optical analysis). This standard analysis by can be easily performed in parts of the collected filter samples. For example, the OC/EC analysis using Sunset analyzer requires just a small punch per filter (usually 1x1 cm or 1x1.5 cm), so sample availability is not an issue. The organic aerosol (OA) concentration can then be calculated by multiplying the measured OC with the OA/OC that is determined from the high-resolution AMS measurements. The product of the factor fractions determined by our proposed method with the OA concentration will then give the mass concentrations of each factor. We have now added a discussion about this important point raised by the reviewer, in the revised paper.

**(2)** Line 31: not quite "automatically", processing is needed. Please remove or revise.

The word "automatically" has been removed.

**(3)** Line 34: change "various sources" to "various sources or atmospheric transformations".

The phrase "various sources" has been replaced by "various sources or atmospheric transformations".

**(4)** Line 34: already defined OA in line 29.

The "organic aerosol (OA)" has been replaced by "OA".

**(5)** Line 41: Regarding, "Its weight and size", add "Its weight, size, and power consumption"

We have rephrased the sentence following the suggestion of the reviewer.

**(6)** Line 74: include sample volume here (flow rate and sample time).

We have added the sampling time and average sampling volume for each campaign in the revised paper.

**(7)** Section 2.1: does the quartz filter survive sonication? Or do they break apart? I'd imagine they break apart, in which case are there not issues with quartz fibers interfering with the atomizer stability if you don't filter the water?

This is a good point. To address this issue we have used the Z-Sizer to measure the particles in water extracts of sonicated clean pre-baked filters and we compared them with those in which we just sonicated clean water. The results showed that no significant difference was observed in the sub-micrometer range. However, in the case of the clean filter, a small peak was observed for particles with sizes around 5 μm. This is probably due to fragments of the quartz filter. However, these larger particles do not make it into the AMS (they do not pass through its aerodynamic lens) and therefore do not affect our measurements. Their presence did not cause any problems in the operation of our atomizer for the hundreds of samples that we have analyzed so far. As a quality assurance measure, we always compare the OA mass spectra in the beginning and in the end of the off-line measurement and we have not seen any change. If something goes wrong with the atomization during a measurement, it would be probably detected this way. We also clean frequently the atomizer (every few samples) to minimize any potential contamination. This would also minimize problems due to the small fragments of the filter. A discussion of these issues has been added to the paper.

**(8)** For blank measurements, has that water gone over a clean filter?

We now clarify that the presented blank measurements refer to atomized clean water. We have also performed measurements for water that has gone over a clean filter using the Z-Sizer. These were the same as that of the clean water for the sub-micrometer particles that are the focus of this work. There were some larger (of the order of 5 μm) particles released from the filter, but these are not measured by the AMS so they do not affect our measurements. Please see also our response to Comment 7 above.

**(9)** Line 93: change "variables related to $CO_2$" to "variables related to $CO_2$ and $CO_2$-based corrections".

We have rephrased the sentence as suggested by the reviewer.

**(10)** Line 96: used the name Thissio campaign here, but previously just referred to it as Athens campaign. Define this as Thissio earlier if that it the name you will use in the document.

To avoid confusion with the two names we refer to the "Athens campaign" everywhere.

**(11)** Section 3: Is there any supporting information that could be provided here. Temperatures during each campaign? Maps of sample sites within the cities/regions? Plots of wind directions/speeds? Something to give the reader a better sense of each study.

We have added the requested supporting information, including the maps of the sampling sites and the average meteorological conditions during each campaign.

**(12)** Line 157: Any ideas on what is driving this difference between BBOA types?
There are a few possible explanations. One is that these factors correspond to fresh and more processed BBOA. The second is that these correspond to different wood types or combustion conditions. This issue clearly deserves additional investigation. We have added a brief discussion of these possible explanation in the revised paper.

**(13)** Figure 2: this time series appears in low resolution on the file I accessed. Also, are the highest concentrations being cut off at the max 10 ug/m3 y-axis scale?
The image resolution has been improved and the scale of the y-axes has been corrected.

**(14)** Also, would be helpful to see a stacked plot time series of the MO-OOA, LO-OOA, and HOA components here. And if showing them in this plot, the MO- and LO-categories should be described in the document when figure first introduced.
A time series plot of the different factors has been added to the supplementary material of the paper.

**(15)** Figure 3: same points as fig. 2, except the resolution does look better for this figure in the file I have. Specify in figure description that these are results from PMF of on-line AMS. Also revise how it is stated in Figure 2. It is not on-line PMF. It is PMF of on-line AMS data.
The description of Figure 3 has been corrected. We now clarify that these results refer the PMF analysis of on-line AMS measurements. The caption of Figure 2 has also been improved.

**(16)** Figure 4: Legend does not indicate which is on-line and which is off-line.
A legend has been added to indicate which is the on-line and which is the off-line. Also this information is now provided in the figure caption.

**(17)** Section 4.1.2.: it would be important to make these online/offline comparisons in terms of mass concentrations, not just percentages of total. It's my understanding that a calibrant was not used for offline samples however.
Indeed, a calibrant was not used for the off-line samples in the current study. Off-line measurements of OC in the same filter samples can be used to convert the fractions calculated here to absolute factor mass concentration in future studies. This is discussed also in our response to comment 1 above. A paragraph has been added to the revised paper to explain this suggested approach for future applications.

**(18)** Line 183: avoid statements such as "The only minor weakness". The truth is it could go either way, maybe it is a weakness, maybe it is a strength, unclear without quantification. Just call it a "difference"
We have followed the suggestion of the reviewer and the phrase "the only minor weakness" has been replaced with "the difference".

**(19)** Line 189: Regarding bootstrapping comparison amongst factors: again, would be better to see this as mass concentrations.

We have added the corresponding information (absolute concentrations) next to the percentages. The absolute concentrations were calculated multiplying the results of the bootstrap analysis with the absolute concentrations of the on-line AMS measurements.

**(20)** Figure S5: This could match better on a mass basis as opposed to a % of total, had a calibrant been used.

We have added in the same figure graphs with the absolute concentrations using as a basis the absolute concentrations of the on-line AMS measurements

**(21)** Line 213: I could imagine a third reason. With high time resolution online AMS, there are more time periods where individual components/factors dominate over the others, producing a more defined factor profile. The offline method always has a higher contribution from all factors due to the daily integration.

We agree with the reviewer, but this is part of our first explanation and it was analyzed in more detail Vasilakopoulou et al. (2022). We have added a couple of sentences here to clarify the point made by the reviewer.

**(22)** It seems there are some major contributions in the offline method from m/z's < 20 relative to the online method. You may get better matches overall if you excluded m/z <20 from your comparison. And in doing so, consider why these may be higher in the offline method (residue from water?)

The reviewer is correct; the water appears to contribute to these differences in the low m/z values. For example, he *m/z* 18 is higher in all off-line spectra compared to the on-line. We have added a brief discussion of this issue in the revised paper.

**(23)** Line 217: Yes, here you accounted for the difference at m/z <20…. why not go back and do the same for factors in previous paragraph (shown in fig S6)

We have followed the suggestion of the reviewer and we have compared the PMF factors between the on-line and the off-line PMF results without accounting for the *m/z's* <20. These comparisons are now discussed in the revised paper.

**(24)** Figure S18: not OOAI and OOAII according to previous figures, MO- and LO-

The legend in the figure has been corrected, and the OOA I and OOA II have been replaced by MO-OOA and LO-OOA.

**(25)** Line 296: fig. S19 not S20

The figure number has been corrected.

**(26)** Line 300: fig. S20 not S21….I think this is off going forward in figures, please fix.

All the figure numbers have been corrected.

**(27)** Line 303: how did authors get O:C from online ACSM? Did it have a ToF-MS?

In order to estimate the O:C from the on-line ACSM measurements we used the approach of Canagaratna et al. (2015) for the unit-mass resolution data. The O:C estimates are clearly more uncertain in this case. This clarification has been added to the paper.

**(28)** Line 335: What do authors mean by "filtration phase"? It was explained that the water extracts were not filtered in their process.
We now clarify in the paper that filtration (1 μm pore size) was used only for these Zeta-Sizer measurements not for the off-line AMS analysis.

**(29)** Line 342: Were the blanks extracted from clean/unused filters, or just looking at the water? There could be suspended particles released from a clean filter.
In this case only the water was used as blank. We have also performed measurements for water that has gone over a clean filter using the Z-Sizer. These were the same as that of the clean water for the sub-micrometer particles that are the focus of this work. There were some larger (of the order of 5 μm) particles released from the filter, but these are not measured by the AMS so they do not affect our measurements. Please see also our response to Comment 7 above. This information has been added to the paper.

**Reviewer 2**

**(1)** This manuscript offers an improved protocol to measure aerosol source apportionment that can be applied to many more areas where online AMS measurement is challenging. The authors provide comprehensive comparison between this improved offline method with corresponding online method very clearly.
We appreciate the helpful suggestions and comments of the reviewer. Our responses (in black) and the corresponding changes in the manuscript follow each comment of the reviewer (in blue).

**(2)** However, more emphasis should be focused on the discussion of the difference, instead of only showing results: in Section 4, multiple paragraphs only report different values and do not report why such variations exist. Meanwhile, it would be helpful if cross-comparison among different campaigns is expanded upon the brief discussion in Section 6. For example, why does summer measurement agree better in offline vs online than winter measurement? Why secondary OOA factors agrees better than HOA and COA?
We have followed the suggestion of the reviewer and have extended the discussion of the differences between on-line and off-line results for the different periods of the year and for the different factors. We believe that the mixing state of the particles (e.g., the co-existence of water-soluble secondary PM components with the water-insoluble primary combustion material) in each period is one important factor affecting the off-line measurements. For example, during the summer the primary particles from transportation are rapidly covered with sulfates and secondary organic aerosol. This process is a slower during the winter, so there are particles that do not include much water soluble material. Our hypothesis is that this water soluble material when present facilitates the transfer of the water insoluble particle core from the filter to the water extract. Concerning the factor differences, secondary OA is much more soluble in water than COA and HOA. This can explain the observed differences between primary and secondary OA. These issues are now explained in the revised paper.

**(3)** On the other hand, there is an absence of evidence on how this improved method out-performs the original method proposed in Daellenbach et al. 2016. It would be helpful if the authors can provide additional evidence and explicit explanation on the results of the original method and improved one.
We should clarify that we do not conclude that the new method outperforms the original method of Daellenbach et al. (2016). Each one of the two methods has its own advantages and drawbacks. The original approach is quantitative, however it requires a specific atomization procedure and does not account directly for the insoluble material. It relies on the use of correction factors that may or may not be applicable for a certain area. The proposed method is easier to apply in different laboratories and the analysis of its results is relatively straightforward because it directly accounts for the insoluble material. This discussion has been added in the revised paper.

**(4)** Line 65: Please provide reasoning behind "significant differences observed in factor spectra" and why the differences matter in the context of offline AMS PMF.

The differences between the on-line and the off-line spectra profiles are a consequence of two facts: the low temporal resolution of the latter have and experimental issues such as the extraction efficiency, etc. We do agree with the reviewer that the differences in the spectra of the factors are of secondary importance for the results of the offline AMS source apportionment. However, it is important to point out that the off-line factor spectra profiles can be quite different from the on-line. A brief discussion about this issue has been added to the revised paper.

**(5)** Line 79-80: "The AMS measures particles smaller than 1 μm. If there are larger particles (e.g. fragments of the quartz filter) in the produced aerosol these do not make it through the AMS aerodynamic lenses" change to "The AMS measures particles smaller than 1 μm because larger particles (e.g. fragments of the quartz filter) in atomization are unable to pass through the AMS aerodynamic lenses". Formality of language.

We have followed the suggestion of the reviewer and we have rephrased this sentence.

**(6)** Line 92-93: Please provide additional information on how you define signals to be weak or bad.

The definition of weak and bad signals has been added to the paper.

**(7)** Line 128: Please be consistent with terminology. PM1 mass concentration in Fig 1 is described as OA concentrations. Either change Fig 1(a) y axis to PM1 mass concentration or change Line 128 to OA concentrations.

We have changed the $PM_1$ to OA in the text of the paper.

**(8)** Line 156 -Line 160: Why separate BBOA into 2 factors and what does each factor represent in the context of BBOA?

The two factors appear in the PMF analysis when 4 or 5 factor solutions are tested. In the 4-factor solution the COA was not present and the four factors were BBOA I, BBOA II, OOA, and HOA, Here, we present the 5 factor solution which also includes the COA factor. The two BBOA factors could correspond to fresh and more processed BBOA or different wood types or combustion conditions. This issue clearly deserves additional investigation. We do agree with the point of the reviewer though that for the purposes of the current work the two BBOA factors could be added to produce one BBOA, since the separation of the two is not clear. We have added a brief discussion of these possible explanations in the revised paper.

**(9)** Line 183- Line 188: When not providing the meaning of BBOA I and BBOA II, it's hard to understand why a difference of ~15% matters.

We do agree that the difference of 15% is not that important. We clarify this issue in the revised paper and we have also added the discussion of the potential meanings of the two biomass burning factors.

**(10)** Line 197-200: Please provide additional information on why the high uncertainty comes from temporal resolution: in Vasilakopoulou et al 2022, it was explained that high uncertainty is a result of challenging separation in low temporal resolution. If this paper holds the same statement, please add it to the explanation.

The differences between the on-line and the off-line results are a consequence of two factors: the low temporal resolution of the latter have and experimental issues such as the extraction efficiency of the filter samples, etc. In Vasilakopoulou et al. (2022) we showed that differences up to 15% can be attributed to the temporal resolution. This explanation and discussion have been added to the paper.

**(11)** In Vasilakopoulou et al 2022, it was shown in high concentration days, each factor has good agreement between 24-hour filter and 30 min filter temporal resolutions, meaning we are not observing a large difference in offline analysis of various temporal resolution. However, this paper shows 24-hour filter still has a large discrepancy when comparing with online 3 min measurement. Does it imply offline measurement in general unable to provide information as accurate as online measurement?

In the analysis of Vasilakopoulou et al. (2022) significant source-apportionment differences were found for individual days. The conclusion of this study was that the source apportionment results were relatively accurate for a month-long period, but the uncertainty was quite high for specific days. We reach the same conclusion for the off-line measurements in this study. The off-line results are quite accurate for a larger period such as one month, but on a daily basis higher discrepancies are observed. A brief discussion of this issue has been added to the paper.

**(12)** Line 208: "Make it to the AMS" should be changed to "detectable by AMS" to be more formal in language use.

We have followed the suggestion of the reviewer and rephrased this sentence.

**(13)** Line 213-216: The difference in mass spec is an interesting observation worth exploring more in detail. For example, Figure S6 suggests offline method tends to underestimate m/z higher than 40 whereas overestimate m/z lower than 40. Is there any scientific hypothesis/publication that implies lower m/z offers higher extraction efficiency, etc. to further expand this paragraph of explanation?

This is a rather complex issue, and the underestimation at $m/z$ higher than 40 is not always present. For example, $m/z$ 43 is overestimated for BBOA I and underestimated for the rest of the factors. Unfortunately, there is not a simple rule that would allow us to connect different $m/z$ ranges with their extraction efficiency.

**(14)** Line 220: Following up on the previous comment, why should we exclude m/z values below 20, other than it shows a large variation between offline and online measurement method?

Since this paper compares the off-line and the on-line measurements we tried to identify the $m/z$ values that cause the differences between the two spectra. In this study we found

that the ones below 20 are responsible for most of the differences observed. We have rephrased this part in the paper suggesting that they should not be excluded, but that they should be checked carefully because they may affect the results.

**(15)** Line 230: Please provide additional information on why we should compare organics to sulfate ratio and what does it imply.

Organics in general, have a wide range of water solubility. They can be either soluble, insoluble or partially soluble in water. Sulfate on the other hand is completely soluble in water for the conditions of the extraction. So comparing the two can give us an estimate of the fraction of the organics that was analyzed with the off-line AMS technique. A discussion of this issue has been added to the paper.

**(16)** Line 270-278: It is good to observe better agreement in this summer campaign as compared to the winter campaign. But can you please add additional scientific discussion on why better agreement is obtained in HOA markers in summer measurement?

This is a good point. As we also explain in our response to Comment 2 of the reviewer, we believe that this behavior is due to the different mixing state of the particles in each period of the year and its effect on the off-line measurements. For example, in summer there is more secondary material as the photochemistry is faster, thus the HOA particles also contain a lot of secondary material. This material is dissolved in the water during the extraction, facilitating the transfer of the remaining insoluble HOA containing particle from the filter to the water extract. This discussion has been added to the revised paper.

**(17)** Line 304-305: Can you please provide O:C information of MO-OOA and LO-OOA factors in the offline analysis? In Vasilakopoulou et al. 2022, LO-OOA always have a lower O:C compared to that of MO-OOA. Such observation in this paper seems odd. If MO-OOA and LO-OOA starts to mix in offline analysis, what is the rationale behind separating the secondary factors into two factors in the first place?

The O:C of the LO-OOA was 0.7 and the O:C of the MO-OOA was 0.86 in the off-line analysis. This is now mentioned in the text and it is consistent with the Vasilakopoulou et al. (2022). Indeed, by definition LO-OOA always has a lower O:C compared to that of MO-OOA. The LO-OOA and MO-OOA separation is performed by the PMF, and they can always be summed to obtain the total OOA. We present and discuss both of them in the present study to facilitate comparisons with the result of the online analysis in the literature.  A brief discussion of this point has been added to the revised paper.

**(18)** Line 306-311: I would hope to see more in-depth discussion on why distribution differs between factors across different campaigns, or if such differences in within uncertainty and acceptable in authors' opinions.

These differences are indeed inside the uncertainty range of the analysis so they do not deserve a detailed discussion.  We clarify this point in the revised paper.

**(19)** Line 310: "LO-OA" should be "LO-OOA".

We have corrected the typo.

**(20)** Line 326-327: The insolubility of HOA and COA is an impressive point to bring up but at the same time, HOA and COA has relatively larger variation in the 3 campaign measurements, compared to other factors. Could this insoluble characteristic explain such large variation?

We agree that the lack of solubility of the HOA and COA components is partially responsible for the variation in the different campaigns. This together with the coating of these particles with soluble secondary material can probably explain the corresponding results. A discussion of this issue has been added to the paper.

**(21)** Section 5.2: It is unclear how this section helps to explain the overhead hypothesis of section 5. I think the authors are trying to imply that some insoluble materials are sent and detected to AMS in measurement step, but it was not emphasized how the results support the hypothesis in this section.

We have rewritten parts of this section to strengthen the links between these results and the overall hypothesis that during the sonication phase the water-soluble material dissolves, facilitating the transfer of the remaining insoluble particle cores from the filter to the water. In order to support our hypothesis, we performed three different tests. First we checked if we can measure suspended particles in water with the Zeta-Sizer (Section 5.1). However, with this test we could not get information about the chemical composition of the particles leaving the filter. So, we moved to the second test (Section 5.2) in which we focused on the least water soluble component of the particles, elemental carbon. The significant reduction of the EC on the filter supports our hypothesis that the insoluble cores leave the filter surface and get suspended in the water. As a third test (Section 5.3) we tested that these insoluble cores can also survive through the atomization and drying processes. These points are now explained better in the revised paper.

**(22)** Line 353-354: How do we know most of the insoluble materials are BC? Or why is BC representative of the insoluble materials?

We do not assume that most of the insoluble material is black carbon, as there are a lot of organic compounds which are partially soluble or totally insoluble in water. We focused on BC because it is water insoluble and easy to measure. We used the BC measurements to prove that the suspended insoluble material can also survive through the atomization and drying processes. A small discussion of this issue has been added to the revised paper.

**(23)** Figure 4: (1) Please provide legends for which bar is online and offline results. (2) This figure will offer more useful information if uncertainty from figure S4 can be incorporated here.

Following the suggestion of the reviewer the figure has been changed, and a legend has been added explaining which bar corresponds to the off-line and which to the on-line results. Also the uncertainty has been incorporated in the graph.

**(24)** Figure 6: I suggest adjusting this figure to the style of Figure 9 where state which plot is CxHy or CxHyO directly on the figure.
Figure 6 has been changed following the suggestion of the reviewer.

**(25)** Figure 8: Can we add uncertainty information onto this figure?
The uncertainty has been added in Figure 8.

**(26)** Figure 9: Inconsistency in figure styles compared to Figure 6, can add the ion fragment formulae to x-axis as Figure 6.
We have redrawn Figure 9 so that it is consistent with Figure 6.

**(27)** Fig S5: This figure can better aid your explanation in Line 197-200 if it can be switched to a diurnal time series plot for each factor comparing online vs offline.
Unfortunately, since the off-line AMS measurements have daily resolution, a diurnal profile cannot be calculated.

**(28)** Figure S8: This figure will be more helpful if you add the total OA concentrations of the 2 days onto (a) and (b) instead of only showing the normalized values.
We have added the total OA concentrations in the figure caption.

**(29)** Figure S13, S14, S20: Figures would be clearer if each factor is color coded as in Figure S6.
Following the suggestion of the reviewer, color has been added to figures S13, S14 and S20.

**(30)** Figure S18: Should OOA I and OOA II correspond to LO-OOA and MO-OOA as Figure S17? Please be consistent with terminology.
The legend in the figure has been corrected, and the OOA I and OOA II have been replaced by MO-OOA and LO-OOA.

**Reviewer 3**

 **(1)** General Comments: In this manuscript, the authors present results from the comparison of three campaigns between offline and online AMS measurements. They find good agreement with the overall PMF characterizations between both methods and suggest that this improvement is due to an increase in the extraction and nebulization of the OA material to include more insoluble material. The paper is well written, and the figures are clear. Overall, the work is a good demonstration of an improved technique. However, there are numerous points where more explanation or clarity is needed. I recommend accepting this work after the following concerns are addressed.
We appreciate the helpful suggestions and comments from the reviewer. Our responses (in black) and the corresponding changes in the manuscript follow each comment of the reviewer (in blue).

Specific comments

**(2)** Please provide more information on the blanking methods that were used throughout the experiments.
We have followed the suggestion of the reviewer and we have added the information concerning the blank measurements. The blank in the off-line experiments is referred to just atomized ultra-pure water.  The same water is used for the extraction of the samples. We have also tested the blank performing the full procedure and a clean filter.

**(3)** On line 81, the blanking looks to be ultra-pure $H_2O$. Was this used to extract blank filters, or is it just a solvent blank? Were any experiments run to test the background with an internal standard, like a labeled salt, included? If the total concentration of the material in the solution is low, the dried particles will be too small to enter the AMS. However, there may still be organic material that is observable when a salt is included to increase the concentration.
The blank used in our study is based on the atomization of the same ultrapure water that is used for the sample extraction. We have also tested the blank performing the full procedure with a clean filter. The results were for all practical purposes the same as those for the clean water (the angle of the spectra was just 3 degrees). This is information is now provided in the revised paper. Please note that particles too small to enter the AMS will necessarily have a very low mass concentration, so even if this material is added to the sample it will not affect our results. We have tested this by just looking at the size distributions of the produced aerosol from the blank experiments with an SMPS.

**(4)** What are the blank spectra for the different campaigns? Were these collected off blank filters? How were the blank filters collected and handled? How many blank filters were collected and run and how much variation was observed?
The blank spectra used for all the corrections were the spectra of the atomized ultrapure water. These blanks were measured during the same days when the actual samples were

analyzed so they account for the small variations in the quality of the ultrapure water. Following the reviewer's suggestion, we have also used the off-line AMS technique for a clean filter. The AMS spectrum was for all practical purposes the same as that of the clean water (angle of 3 degrees between the spectra). While the presence of the filter did not make any difference in the sub-micrometer particle range analyzed by the AMS a small peak was observed in the Zeta-Sizer measurements for particles with sizes around 5 μm. This is probably due to fragments of the quartz filter. However, these larger particles do not make it into the AMS (they do not pass through its aerodynamic lens) and therefore do not affect our measurements. This information has been added to the paper.

**(5)** In the paragraph under 2.3 Off-line source apportionment the subtraction of blanks is mentioned. How was blank subtraction carried out for the offline analysis? Was this done prior to PMF analysis?
The blank subtraction was performed prior to the PMF analysis. Both the off-line spectra and errors were corrected for the blank measurements. More specifically, the blank concentration was subtracted from the sample for each *m/z* separately. This is now clarified in the revised paper.

**(6)** In section 5.1, is the blank that is used for comparison an extraction of a blank filter?
The blank that is used in our standard procedure is that of atomized ultrapure water. The blank from the extraction of a clean filter is for all practical purposes the same as that of ultrapure water in our approach (the angle of the AMS spectra was just 3 degrees). This is now mentioned in the revised paper.

**(7)** In section 4.1.1 two BBOA factors were observed, but the mass spectra and time series are very similar. Why was a solution with two BBOA selected?
The two BBOA factors appear in the PMF analysis when 4 or 5 factor solutions are tested. In the 4-factor solution the COA was not present and the four factors were BBOA I, BBOA II, OOA and HOA. Here, we present the 5 factor solution which also includes the COA factor. The two BBOA factors could correspond to fresh and more processed BBOA or different wood types or combustion conditions. This issue clearly deserves additional investigation. For the purposes of the current work the two BBOA factors could be added to produce one BBOA, since the separation of the two is not clear. We have added a brief discussion of these possible explanations in the revised paper.

**(8)** In all the diurnal patterns in the supplemental, what was the variation throughout the day? I often see this as a shaded region. Please include something like this or another way to communicate the variability.
We have followed the suggestion of the reviewer and added the standard deviation of the mean as a shaded region in the corresponding diurnal profiles.

**(9)** The numbering for the figures is confusing because Figures 2 and 3 are not discussed until much later in the manuscript. I recommend renumbering the figures so that they are listed in the order they are mentioned in the text.

We have followed the suggestion of the reviewer and Figure 2 is now mentioned in Section 3.2, while Figure 3 is first mentioned in Section 3.3.

**(10)** In Figure 4 there is no label for offline vs. online, please add that. Also, bootstrapping analysis is carried out for all the offline work. Can you include these results as error bars on the figures?

Following the suggestion of the reviewer, the figure has been changed, and a legend has been added explaining which bar corresponds to the off-line and which to the on-line results. Also the uncertainties have been added to the graph.

**(11)** On line 200 it is mentioned that the uncertainty on a daily basis can be really high and that this is due to the temporal resolution of the offline measurements with a citation. Please provide more text here to describe what is meant by this. Also, this suggests that the uncertainty on a daily level might decrease if the same time resolution is used for the online measurements. Essentially downgrading the time resolution. Was something like this done? That would be a more direct comparison for this portion of the analysis.

In Vasilakopoulou et al. (2022) we quantified the effect of the temporal resolution in the PMF analysis of on-line AMS measurements. More specifically, a five-month period of on-line ASCM measurements with temporal resolution of 30 min was used for the study. The 30 min were averaged to daily resolution and the daily PMF results were compared to the 30 min. The average contribution to the total OA of each factor varied within 8% between the lowest and the highest temporal resolution results for the five-month study, something that is quite encouraging for the use of off-line AMS measurements. However, on a daily basis discrepancies of a factor of 2 were observed for several factors. when the results of the 30 min and 24 h analyses were compared. For this comparison the 30 min PMF results were averaged to daily resolution and compared to the low-resolution PMF results. This suggests that the offline AMS results can be quite uncertain for certain days. A brief summary of the conclusions of our previous work has been added to the revised paper.

**(12)** On line 215 a few different reasons for the differences are listed. What was the particle size range that was collected for each filter? Were cyclones included and if so, what was the cut-point? What was the photochemistry like during the collection? Might there be some additional aging of the material on the filter during collection? What about volatilization of semi-volatile organic compounds off the filter? These are a few other reasons I can think of that might contribute to the differences observed between the offline and online measurements and I would recommend including at least some of them in the discussion.

$PM_{2.5}$ cyclones were used for the filter sampling. After the collection of each filter, the samples were wrapped in pre-baked aluminum foil and were placed in petri dishes.

Subsequently they were placed in a freezer at -18°C. Also, the samples were transferred in boxes with ice cubes to the laboratory. In this way we tried to reduce as much as possible the volatilization of semi-volatile organic compounds of the filter. We have added a discussion about these issues that could contribute to the differences between the on and off-line AMS results in the revised paper.

**(13)** On lines 221-223 there is a discussion of the offline method capturing 64% of the $C_xH_y^+$ and 82% of the $C_xH_yO^+$. I do not understand what is being communicated here (and in the later portions of this text where this same analysis is carried out). What does it mean that these percentages are being captured?

In this section we attempt to use the high-resolution AMS results to gain further insights into the similarities and differences of the spectra in the approaches (on and off-line). The $C_xH_yO^+$ fragments contain oxygen so they should be coming from relatively water soluble compounds. The difference in the two approaches is 18%, suggesting that indeed the off-line AMS approach captures these compounds to a very large extent. The $C_xH_y^+$ fragments are characteristic of the hydrocarbon-like OA (HOA) that has in general low water solubility. The fact that 64% of the material is present in the off-line organic mass spectra provides strong evidence that our approach extracts and sends for AMS analysis the majority of these mostly water insoluble compounds. If these compounds remained on the filter there would be no signal at the corresponding m/z values for $C_xH_y^+$. A discussion about that is added in the revised paper.

**(14)** In section 4.2.1 the difference between the MO-OOA and LO-OOA factors is noted to be small and the spectra look very similar. For the offline analysis however, they are larger. How does this compare to the differences that are observed in other campaigns between these secondary factors?

In the PMF analysis of the on-line measurements the O:C of the MO-OOA was 1 and LO-OOA O:C was 0.94. Their angle was 9°. In the PMF analysis of the off-line measurements the angle between the MO-OOA and LO-OOA spectra 35$^0$. The O:C of the MO-OOA was 1.2 and of the LO-OOA 0.81 in this case. These changes in the spectra of the factors appear to be characteristic of the low-temporal resolution off-line analysis. The differences of the LO-OOA and MO-OOA spectra vary widely in different campaigns because they appear to represent the upper and lower limits of SOA oxidation encountered in the specific campaign. These differences are discussed in the revised paper.

**(15)** There are some errors in the numbering of the supplemental figures in section 4.3.2, please correct that.

The numbering of the figures in the SI has been corrected in the revised paper.